# Mitochondrial respiratory function is preserved under cysteine starvation via glutathione catabolism in NSCLC

Nathan P. Ward [1], Sang Jun Yoon[1], Tyce Flynn[1], Amanda M. Sherwood[1], Maddison A. Olley[1], Juliana Madej[1] & Gina M. DeNicola [1]

Cysteine metabolism occurs across cellular compartments to support diverse biological functions and prevent the induction of ferroptosis. Though the disruption of cytosolic cysteine metabolism is implicated in this form of cell death, it is unknown whether the substantial cysteine metabolism resident within the mitochondria is similarly pertinent to ferroptosis. Here, we show that despite the rapid depletion of intracellular cysteine upon loss of extracellular cystine, cysteine-dependent synthesis of Fe-S clusters persists in the mitochondria of lung cancer cells. This promotes a retention of respiratory function and a maintenance of the mitochondrial redox state. Under these limiting conditions, we find that glutathione catabolism by CHAC1 supports the mitochondrial cysteine pool to sustain the function of the Fe-S proteins critical to oxidative metabolism. We find that disrupting Fe-S cluster synthesis under cysteine restriction protects against the induction of ferroptosis, suggesting that the preservation of mitochondrial function is antagonistic to survival under starved conditions. Overall, our findings implicate mitochondrial cysteine metabolism in the induction of ferroptosis and reveal a mechanism of mitochondrial resilience in response to nutrient stress.

The proteinogenic amino acid cysteine (Cys) is a significant contributor to cellular functions due to its resident thiol group. Cysteine supplies this reactive sulfur in support of cellular antioxidant capacity (glutathione [GSH]), electron transfer (iron-sulfur [Fe-S] clusters), energy metabolism (coenzyme A), and osmoregulation (taurine)[1]. Despite the essentiality of cysteine for these processes, there is an apparent therapeutic window in restricting cysteine systemically as an anti-cancer strategy[2,3]. Indeed, cysteine becomes conditionally essential across many tumor species[4], in part due to an inability to synthesize this amino acid from methionine through the transsulfuration pathway[5]. This dictates a reliance on an extracellular source of cysteine, predominantly in its oxidized form of cystine ($Cys_2$). Cysteine availability can be targeted through its systemic depletion[2] or inhibition of the primary $Cys_2$ transporter, system $X_c^-$[6].

Cysteine restriction promotes an iron-dependent form of cell death known as ferroptosis[7], where Fenton chemistry promotes the generation of free radicals that initiate lipid radical formation and the unsustainable peroxidation of plasma membrane phospholipids[8]. The resulting loss of membrane integrity is suppressed by intrinsic pathways mediated by ferroptosis suppressor protein 1 (FSP1)[9,10] and glutathione peroxidase 4 (GPX4)[11,12] to ward off ferroptosis under basal conditions. However, in the absence of cysteine, cancer cells are unable to synthesize GSH, thereby restricting GPX4 activity in preventing ferroptosis.

Additional factors have been implicated in the induction of ferroptosis[13], including the engagement of mitochondrial metabolism and the tricarboxylic acid (TCA) cycle[14,15]. Mechanisms by which the TCA cycle promotes ferroptosis include the generation and export of citrate to drive the synthesis of new substrates for lipid peroxidation[16], and the stimulation of the electron transport chain (ETC)[15]. However, how ETC activity promotes the induction of ferroptosis remains to be fully elucidated[17]. Existing evidence suggests that ATP production[16]

[1]Department of Metabolism & Physiology, Moffitt Cancer Center, Tampa, FL, USA. ✉e-mail: nathan.ward@moffitt.org; gina.denicola@moffitt.org

and the generation and extramitochondrial release of reactive oxygen species (ROS) may be relevant consequences[17,18].

One aspect of mitochondrial metabolism that has not been adequately addressed is the role of resident cysteine metabolism within the mitochondria. Though the disruption of cytosolic cysteine metabolism (i.e., GSH synthesis) plays a significant role in the induction of ferroptosis under cysteine limitation[5,19], it remains unknown if alterations in mitochondrial cysteine metabolism are also operative. Mitochondrial cysteine is used in the synthesis of mitochondrially encoded protein components of the ETC, hydrogen sulfide production[20], and Fe-S cluster synthesis[21] in support of bioenergetic metabolism. Given the established connection between oxidative metabolism and ferroptosis[14,15], the requirement for cysteine in support of respiration suggests it may be particularly relevant to ferroptosis. Furthermore, a disruption in mitochondrial cysteine metabolism could explain the gradual loss of mitochondrial function reportedly associated with prolonged cysteine deprivation[3,15,22,23].

To interrogate the role of mitochondrial cysteine in Fe-S cluster synthesis during ferroptosis, we evaluated the influence of prolonged cysteine deprivation on mitochondrial respiratory function. We evaluated this in the context of non-small cell lung cancer (NSCLC), given its sensitivity to cysteine starvation[24] and the robust mitochondrial metabolism exhibited by human lung tumors[25,26]. Remarkably, despite the rapid depletion of mitochondrial cysteine under starved conditions, Fe-S cluster synthesis persisted in support of respiratory function through the onset of ferroptosis. This was achieved through the catabolism of GSH to mobilize cysteine stored within this tripeptide at the detriment of NSCLC viability.

## Results

### Mitochondrial respiratory function persists under cystine deprivation

To determine the influence of cysteine deprivation on mitochondrial metabolism, we restricted a panel of NSCLC cells of extracellular cystine. We opted to starve cells of their source of cysteine rather than inhibit cystine uptake with erastin to directly assess the consequences of cysteine deprivation without the confounding influence of erastin's promiscuous inhibition of the mitochondrial voltage-dependent anion channel[27,28]. As previously described, NSCLC cells succumbed to ferroptosis in the absence of cysteine[5,29] (Supplementary Fig. 1a–c). We began to observe the accumulation of ferroptotic cells, as detected upon the intercalation of a cell-impermeable fluorescent dye within nuclear DNA, following 24 h of cystine deprivation (Supplementary Fig. 1a). Therefore, we performed our mitochondrial analyses upon 20 h of starvation to ensure sufficient nutrient stress in the absence of a loss of cell viability.

To assess general mitochondrial respiratory function under cystine deprivation, we employed the Seahorse MitoStress test. Generally, the basal oxygen consumption rate (OCR) of NSCLC cells was not diminished in the absence of cystine (Fig. 1a), indicating a retention of mitochondrial respiration. This coincided with a maintenance of mitochondrial coupling efficiency (Fig. 1b), an indicator of the functionality of the mitochondrial inner membrane (IMM)[30]. Further, though it was reported that the IMM becomes dramatically hyperpolarized under cysteine deprivation[15], mitochondrial membrane potential ($\Delta\Psi_m$) was either minimally increased or unchanged under starved conditions (Fig. 1c). Consistent with the preservation of a functional IMM, ATP-linked respiration was not compromised in the absence of cystine (Fig. 1d). Moreover, several NSCLC lines exhibited a significant elevation in maximal respiratory capacity that could not be explained by an increase in mitochondrial density (Fig. 1e, f), suggesting that cysteine deprivation may enhance mitochondrial function beyond the requirement for oxidative phosphorylation. In contrast, mitochondrial function was diminished by cystine deprivation in A549 cells. Interestingly, acute supplementation with cystine restored respiratory

function in these cells, suggesting cysteine may act as a respiratory substrate (Supplementary Fig. 1d, e). Cysteine catabolism produces both pyruvate and hydrogen sulfide ($H_2S$), which can directly donate electrons to the electron transport chain[20,31]. Therefore, A549 cells may be more reliant on cysteine to drive respiration. Still, these data broadly indicate a robust persistence of mitochondrial respiration under cystine deprivation that occurs independent of an expected insult to the IMM[15,22].

### Cystine deprivation does not promote mitochondrial oxidative stress

To confirm that the persistence of respiratory function in NSCLC was not associated with mitochondrial stress as has been previously reported in other contexts[3,15,22,23], we interrogated the mitochondrial redox state in response to cystine starvation. First, we employed a series of fluorescent ROS indicators to assess changes in the mediators of oxidative stress (Supplementary Fig. 1f–j). While we observed the hallmark increase in lipid peroxidation at the cellular level (Fig. 2a)[8], and a coincident increase in cellular free radical levels under cystine deprivation (Fig. 2b), these indications of oxidative stress did not extend to the mitochondria. Use of specific and mitochondrially targeted indicators of superoxide ($\cdot O_2^-$) and hydrogen peroxide ($H_2O_2$) revealed that prolonged cystine starvation did not alter the abundance of either species (Fig. 2c, d). This suggests that cysteine deprived mitochondria can scavenge the significant ROS generated as a byproduct of persistent mitochondrial metabolism[32].

Mitochondrial ROS metabolism is achieved predominantly through the activity of two complementary, yet distinct, thiol-based antioxidant systems[33]. Both the GSH tripeptide and the small molecular weight protein thioredoxin (TXN) support ROS detoxification by a network of antioxidant proteins within the mitochondria[34]. Through the use of a genetically encoded and mitochondrially targeted biosensor[35], we determined that the mitochondrial GSH pool was more reduced under cystine deprivation relative to replete conditions (Fig. 2e). Further, an assessment of the oxidation state of peroxiredoxin 3 (PRDX3), a component of the TXN system, revealed an increase in the functional reduced form upon cystine starvation (Fig. 2f). Together these data suggest that cystine deprivation elicits a more reduced mitochondrial matrix despite robust metabolic activity. In agreement with this apparent lack of oxidative stress, we did not observe mitochondrial lipid peroxidation upon cystine deprivation in the majority of our cell lines (Fig. 2g), which indicates the increase in whole cell lipid peroxidation is confined to the plasma membrane (Fig. 2a). Again, A549 cells were an outlier as they were the only cell line examined to exhibit significant elevations in mitochondrial $H_2O_2$ and lipid peroxidation (Fig. 2d, g), which suggests that the mitochondria of these particular cells are far less insulated from the pro-ferroptotic oxidative stress induced by cystine starvation (Fig. 2a). Nonetheless, these data reflect that a maintenance of mitochondrial redox homeostasis accompanies the persistence of respiration in cysteine restricted NSCLC cells.

### Mitochondrial Fe-S cluster synthesis is sustained under cystine deprivation

Cysteine is a requirement for mitochondrial function, due in large part to its utilization in the synthesis of Fe-S clusters. These redox cofactors mediate electron transfer and support the enzymatic function of proteins critical to mitochondrial metabolism, including components of the ETC and TCA cycle[21]. Fe-S cluster synthesis is initiated in the mitochondria and requires the coordination of iron and cysteine-derived sulfur by a multi-protein complex (Fig. 3a). Disrupting this machinery is associated with the loss of ETC function and an increase in ROS production[36,37]. Furthermore, deficiencies in Fe-S cluster synthesis are associated with severe mitochondrial defects underlying varied human pathologies[38]. Considering this, the absence of

mitochondrial dysfunction in NSCLC cells under cystine starvation suggests that Fe-S cluster synthesis is sustained.

To scrutinize this possibility, we evaluated the functionality of mitochondrial Fe-S proteins under cystine deprivation. First, an assessment of the TCA cycle enzyme aconitase (ACO2) in isolated mitochondria showed no change in ACO2 activity upon cystine starvation (Fig. 3b). Next, we employed a specialized Seahorse-based assay to specifically assess the function of the Fe-S cluster dependent complexes of the ETC[39,40]. This revealed that the activities of the respiratory supercomplexes I-III and II-III were not diminished in the absence of extracellular cystine (Fig. 3c, d). Lastly, we interrogated the function of the unique Fe-S protein lipoic acid synthetase (LIAS). Rather than simply mediate its catalysis, the resident Fe-S cluster of LIAS is consumed during the synthesis of lipoate moieties that facilitate metabolic protein function[41]. We evaluated the lipoylation status of the E2

component of the pyruvate dehydrogenase complex (PDH-E2) and dihydrolipoamide S-succinyltransferase (DLST) as surrogate markers for LIAS activity and found no change in protein lipoylation with respect to cysteine availability (Fig. 3e). Collectively these data demonstrate a complete maintenance of Fe-S protein function up to 20 h of cystine starvation, suggesting that Fe-S cluster synthesis is sustained despite the absence of cysteine.

Still, there remained the possibility that Fe-S cluster turnover exceeds the 20 h starvation timeframe used for these studies, which would negate the necessity of de novo synthesis. To interrogate this, we temporally assessed the function of Fe-S proteins in response to various insults to the Fe-S cluster synthesis pathway. In addition to limiting sulfur through starving cells of cystine, we restricted iron availability by chelating cellular iron with deferoxamine (DFO)[41] or ablated synthesis altogether by silencing expression of the vital Fe-S biosynthetic

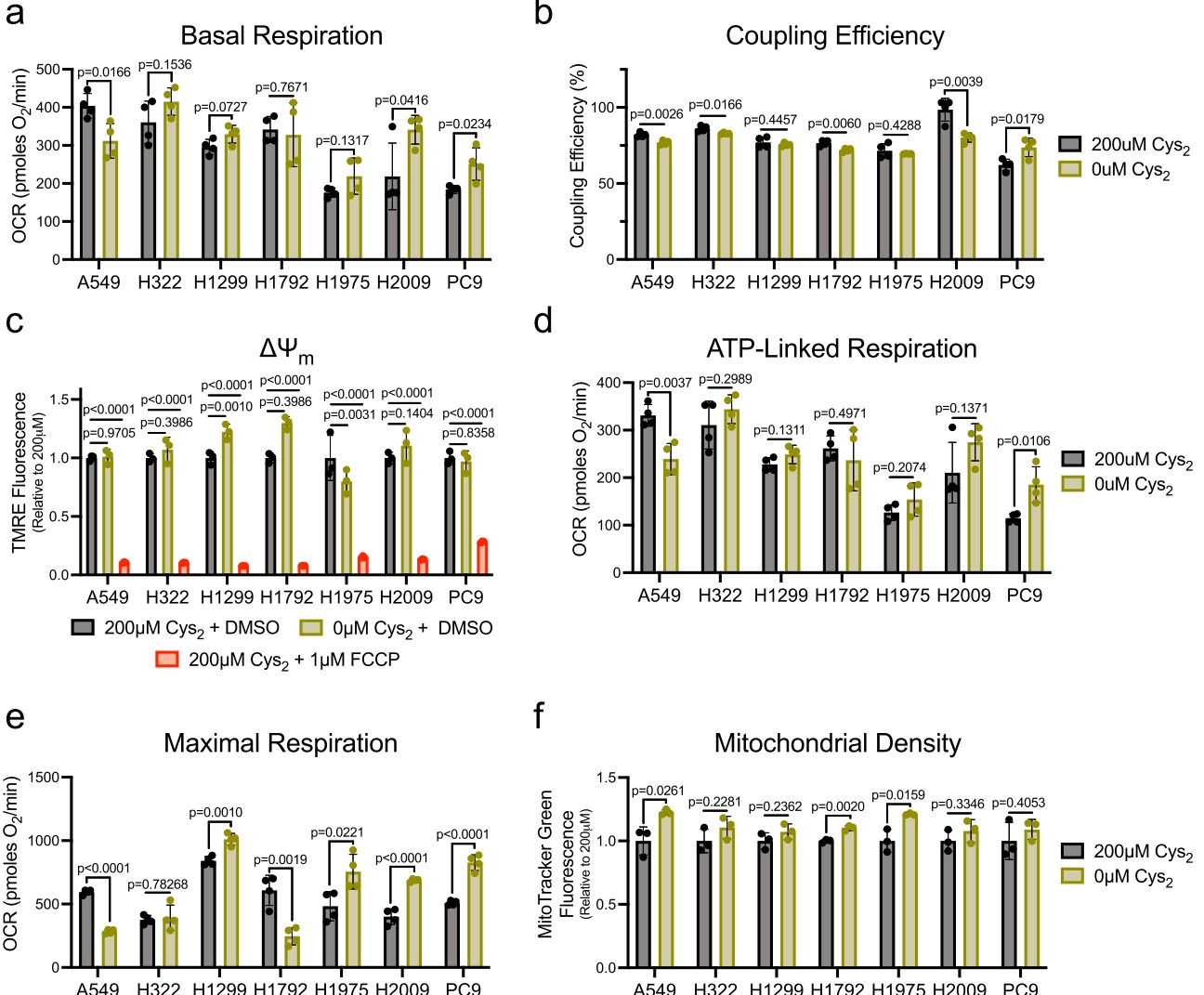

**Fig. 1 | Mitochondrial respiration is sustained in the absence of extracellular Cys₂. a** Basal oxygen consumption rate of NSCLC cells stimulated with 10 mM glucose and 2mM L-glutamine following culture in the presence or absence of cystine ($n = 4$ biologically independent samples per condition). **b** Percentage of basal oxygen consumption linked to the generation of ATP in NSCLC cells fed or starved of cystine ($n = 4$ biologically independent samples per condition).
**c** Average relative TMRE fluorescence of NSCLC cells cultured with indicated [cystine] or treated with FCCP for 15 min ($n = 3$ biologically independent samples per condition). **d** Oligomycin-sensitive oxygen consumption rate of cystine-fed or starved NSCLC cells ($n = 4$ biologically independent samples per condition).

**e** FCCP-stimulated oxygen consumption rate in in NSCLC cells fed or starved of cystine ($n = 4$ biologically independent samples per condition). **f** Average relative MitoTracker Green fluorescence of NSCLC cells cultured in the presence or absence of cystine ($n = 3$ biologically independent samples per condition). For all experiments, cells were cultured in the indicated [cystine] for 20 h prior to the indicated analysis. Data represent mean values ± s.d. For (**a**, **b**, **d–f**), P values were calculated using a two-tailed unpaired Student's *t*-test. For (**c**), P values were calculated using a one-way ANOVA. All data are representative of at least 3 experimental replicates. For (**a–f**), source data provided as a Source Data file.

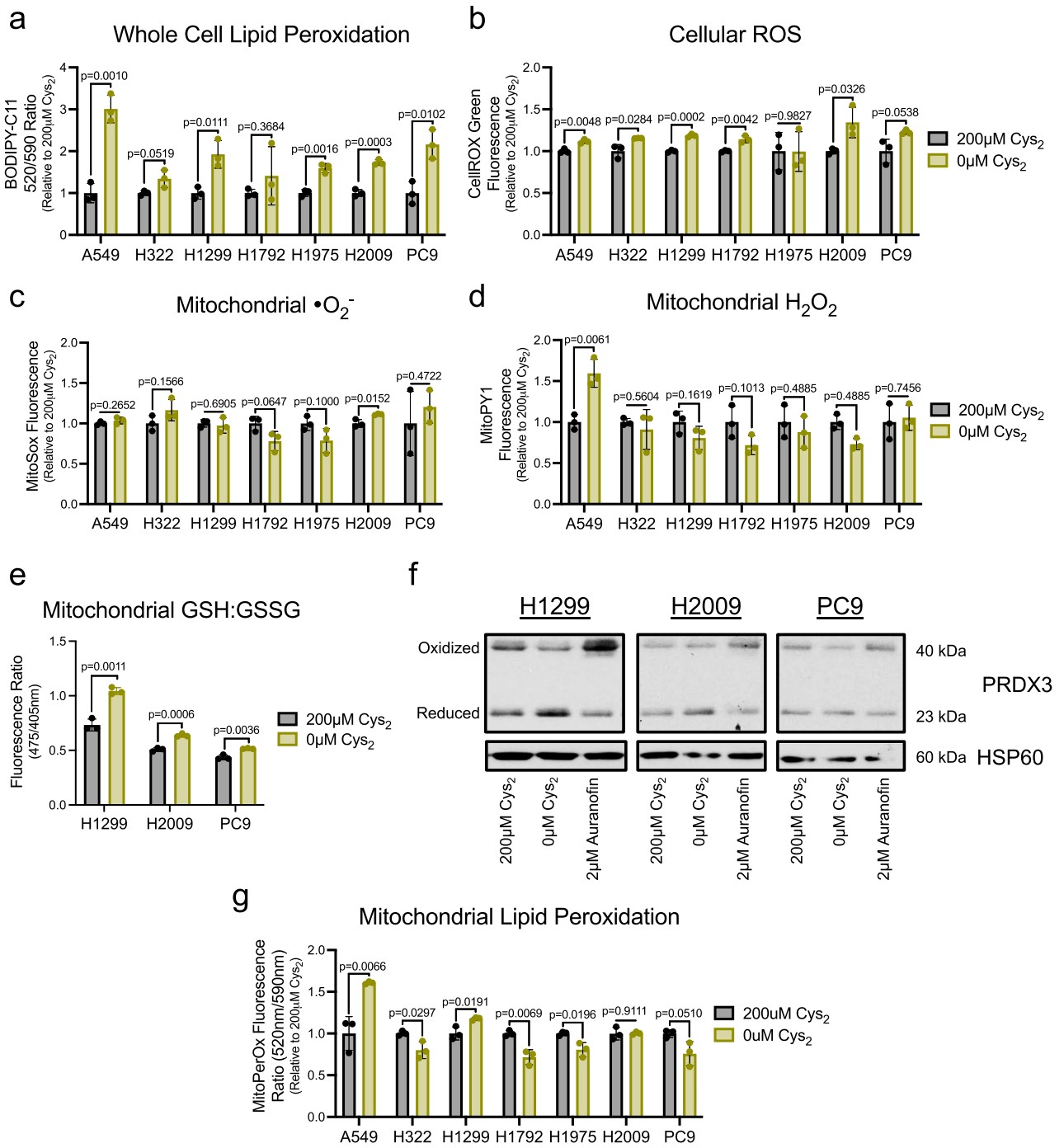

**Fig. 2 | Cys₂ starvation does not promote mitochondrial oxidative stress.** Following culture in the presence or absence of cystine, NSCLC cells were treated with ROS-sensitive fluorescent dyes to determine the average relative levels of (**a**) cellular membrane lipid peroxides, (**b**) cellular ROS, (**c**) mitochondrial superoxide, or (**d**) mitochondrial hydrogen peroxide ($n = 3$ biologically independent samples per condition for each experiment). **e** Ratio of Mito-Grx1-roGFP2 fluorescence as indicative of the proportion of the biosensor bound to reduced (GSH) or oxidized (GSSG) glutathione in H1299, H2009, and PC9 cells cultured with or without cystine for 16 h ($n = 3$ biologically independent samples per condition). **f** Redox immunoblotting of HSP60 and the PRDX3 oxidation state in H1299, H2009, and PC9 cells following culture with the indicated [cystine] or 1 h treatment with auranofin. **g** Relative MitoPerOX fluorescence ratio of NSCLC cells cultured in the presence or absence of cystine to indicate the extent of mitochondrial membrane lipid peroxidation ($n = 3$ biologically independent samples per condition). For (**a–d**, **f**, **g**) cells were cultured in the indicated [cystine] for 20 h prior to the indicated analysis. Data represent mean values ± s.d. For (**a–e**, **g**) $P$ values were calculated using two-tailed unpaired Student's $t$-test. All data are representative of at least 3 experimental replicates. For (**a–g**), source data provided as a Source Data file.

machinery components, NFS1 and ISCU. Upon establishing the kinetics of shRNA-mediated knockdown of these proteins in H1299 cells, we determined the onset of protein depletion to be 48 h post-lentiviral infection (time-point 0) (Fig. 3f and Supplementary Fig. 2a). We then compared the activities of ACO2 and the Fe-S cluster dependent

respiratory complexes and observed that in contrast to cystine starvation, the functions of these proteins decay within 16 h of DFO treatment or NFS1 or ISCU knockdown (Fig. 3g–i). This decay in protein function in response to DFO treatment or NFS1 or ISCU knockdown corresponded with a gradual loss of mitochondrial Fe-S protein expression (Fig. 3j and

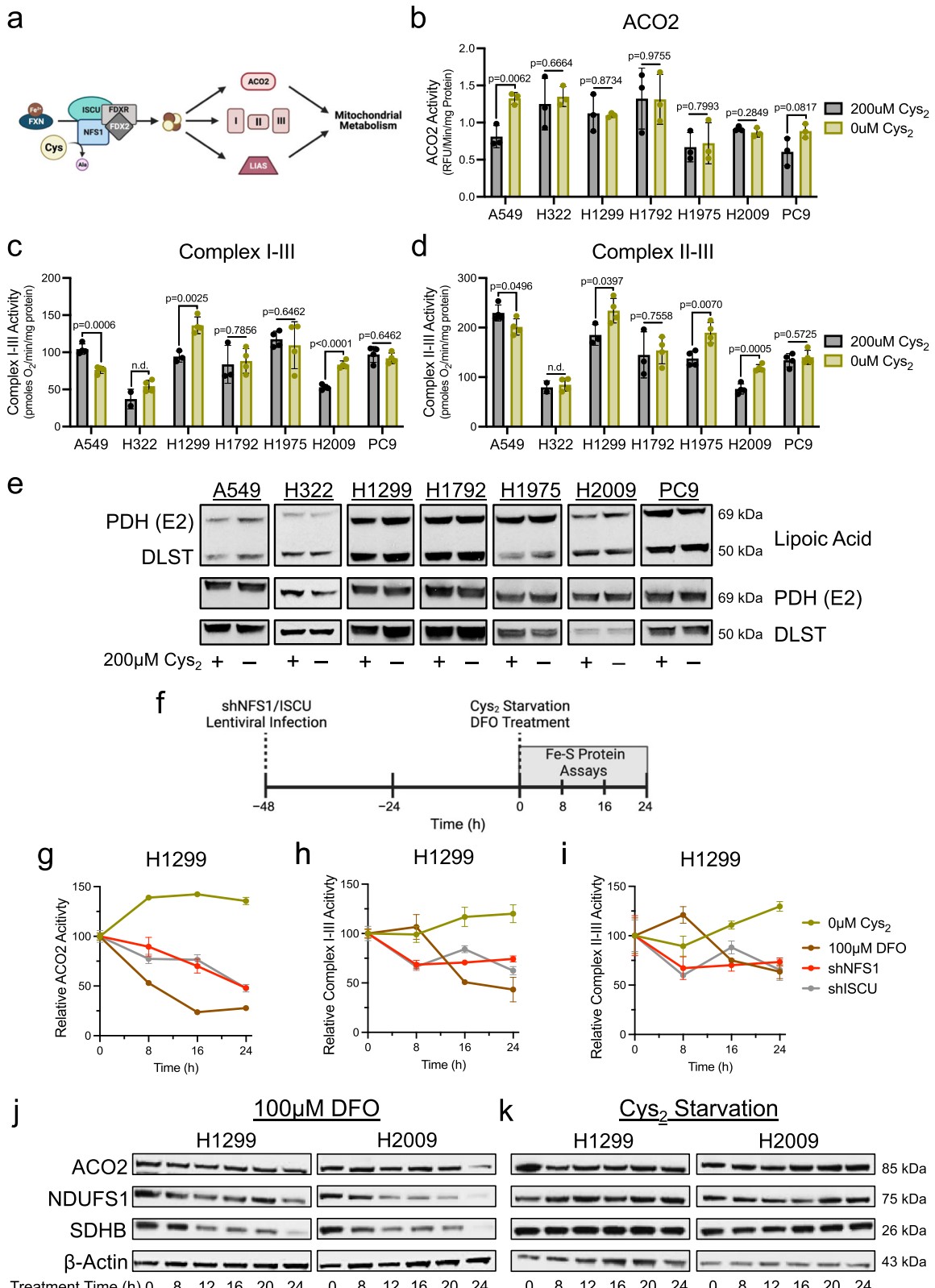

Supplementary Fig. 2b), consistent with the association between the stability of these proteins and the fidelity of Fe-S cluster synthesis[41]. Importantly, the progressive decrease in Fe-S protein function and expression with DFO treatment (Fig. 3g–j) suggests that rather than stripping iron from existing clusters and rendering them inactive, iron chelation restricts synthesis of new clusters. Therefore, these data indicate a need for de novo synthesis within the time frame of our

analyses (Fig. 3b–e). In addition to the retention of mitochondrial Fe-S protein function under cystine starvation, we also find that protein expression is sustained during this time frame (Fig. 3k), further indicating a resilience of this pathway in response to cysteine restriction. Interestingly, the stability of extramitochondrial Fe-S proteins was more varied in response to cystine starvation, with some proteins displaying reduced expression within 20 h of starvation, while the expression of

**Fig. 3 | Fe-S protein function is preserved under Cys₂ deprivation. a** Schematic of Fe-S cluster synthesis and its contribution to enzymatic function that supports mitochondrial metabolism; Ala, alanine, Cys, cysteine, Fe²⁺, iron, I, complex I, II, complex II, III, complex III. **b** ACO2 activity in mitochondrial lysates of NSCLC cells fed or starved of cystine ($n = 3$ biologically independent samples per condition). **c** Supercomplex I-III activity in permeabilized NSCLC cells following culture in the presence or absence of cystine ($n = 4$ biologically independent samples per condition, except when $n = 2$ for H322 + 200 μM and $n = 3$ for H1299 and H1792 + 200 μM). **d** Supercomplex II-III activity in permeabilized NSCLC cells following culture in the presence or absence of cystine ($n = 4$ biologically independent samples per condition, except when $n = 2$ for H322 + 200 μM and $n = 3$ for H1299 and H1792 + 200 μM). **e** Immunoblot analysis of PDH complex subunit E2 and α-ketoglutarate dehydrogenase subunit DLST lipoylation following culture in cystine replete or starved conditions. **f** Experimental timeline for the assessment of Fe-S proteins in response to insults to Fe-S cluster synthesis (**g**–**i**). **g** ACO2 activity in mitochondrial lysates of H1299 cells following treatment with cystine starvation, DFO, or upon knockdown of NFS1 or ISCU at the indicated time points ($n = 3$ biologically independent samples per condition). **h, i** ETC supercomplex activities in permeabilized H1299 cells following treatment with cystine starvation, DFO, or upon knockdown of NFS1 or ISCU at the indicated time points ($n = 3$ biologically independent samples per condition). For (**g**–**i**), data points are representative of the activity of each Fe-S protein relative to the activity of that protein at time 0 h. Immunoblot analyses of mitochondrial Fe-S proteins (ACO2, NDUFS1, and SDHB) in H1299 and H2009 cells following culture in (**j**), cystine-deficient media or (**k**), DFO for the indicated time. For (**g**–**i**), all cells were cultured in the presence of 1 μM ferrostatin-1. Data represent mean values ± s.d; n.d. not determined. For (**b**–**d**) $P$ values were calculated using two-tailed unpaired Student's $t$-test. All data are representative of at least 3 experimental replicates. **a, f** created with BioRender.com released under a Creative Commons Attribution-NonCommercial-NoDerivs 4.0 International license, https://creativecommons.org/licenses/by-nc-nd/4.0/deed.en. For (**b**–**e**) and (**g**–**k**), source data provided as a Source Data file.

others was unchanged (Supplementary Fig. 2c). Like ACO2, activity of the cytosolic aconitase (ACO1) was far more resistant to cysteine limitation relative to other insults to Fe-S cluster synthesis (Supplementary Fig. 2d). To determine the point at which mitochondrial Fe-S protein function is compromised by cystine starvation, we assessed respiratory complex activity in cells subject to cystine deprivation for 0–96 h that were concomitantly treated with ferrostatin-1 to prevent the induction of ferroptosis. We found that ETC supercomplex activities begin to progressively decline by 48 h of starvation (Supplementary Fig. 2e), suggesting there is a point at which sulfur becomes limiting. Together, these data indicate that NSCLC cells retain the capacity to synthesize Fe-S clusters within the mitochondria in the absence of an extracellular cysteine source.

## CHAC1 supports Fe-S protein function under cystine deprivation

The maintenance of Fe-S cluster synthesis under cystine starvation argues for the existence of an alternative source of mitochondrial cysteine. Given the absence of mitochondrial oxidative stress and more reduced state of the matrix under cystine starvation (Fig. 2), we considered the possibility that the antioxidant function of GSH was dispensable, and that this tripeptide could serve as a cysteine sink to support the mitochondrial cysteine pool. Intriguingly, characterization of the transcriptional response to cysteine deprivation has revealed glutathione-specific gamma-glutamylcyclotransferase 1 (CHAC1) as the most highly induced gene[6]. CHAC1 is a component of the intracellular GSH cleavage system (Fig. 4a), which catabolizes GSH to yield 5-oxoproline and cysteinylglycine (Cys-Gly)[42]. Cys-Gly can then be further cleaved by various peptidases to release its constituent amino acids[43]. Considering this, we hypothesized that CHAC1 induction under cystine starvation would promote cysteine mobilization from GSH to support Fe-S cluster synthesis.

In our NSCLC cells, we find that cystine starvation elicited a temporal accumulation of CHAC1 protein that coincides with an induction of ATF4 (Fig. 4b), agreeing with previous reports that CHAC1 is an ATF4 target[42,44]. In contrast, expression of CHAC2, the other intracellular mediator of GSH catabolism, was not induced by cystine starvation (Fig. 4b). Several expansive studies mapping the subcellular distribution of the human proteome have suggested that CHAC1 localizes to multiple organelles, including the mitochondria[45,46]. We generated cells expressing a His-tagged construct within the outer mitochondrial membrane (HA-Mito)[47] to facilitate the purification of mitochondria from cell homogenates and assess CHAC1 mitochondrial localization. We found that CHAC1 is expressed within the mitochondrial compartment of NSCLC cells and that mitochondrial CHAC1 expression was also increased under cystine starvation (Fig. 4c). Importantly, CHAC2 did not localize to the mitochondria irrespective of extracellular cystine availability, further suggesting divergent roles for these proteins in mediating intracellular GSH catabolism.

Because CHAC1 is not specifically confined to a single compartment (Fig. 4c), we performed immunofluorescence to validate the association between CHAC1 and the mitochondria. In agreement with our immunoblotting analysis, we observed an induction of CHAC1 under cystine starvation in CHAC1 expressing H1299 cells but not those subject to CRISPR/Cas9-mediated CHAC1 knockout[48], indicating antibody specificity (Supplementary Fig. 3a–c). Though we observed diffuse expression of CHAC1 throughout the cell, we did detect colocalization between a fraction of CHAC1 and the mitochondria. Importantly, this was enhanced under cystine starvation (Supplementary Fig. 3d, e). Similarly, enhanced CHAC1 expression and association with the mitochondria were observed in H2009 cells (Supplementary Fig. 3f–h). Given that *CHAC1* does not encode a canonical mitochondrial localization sequence, we next sought to identify the sub-organellar localization of CHAC1 within the mitochondria. Mitochondrial isolates from NSCLC cells were incubated with the indiscriminate protease, proteinase K, in the presence or absence of detergent to differentiate matrix proteins from those external to the IMM[49]. Compared to matrix proteins, which were largely insulated from degradation in the absence of detergent, or the outer mitochondrial membrane protein TOM20, which was fully degraded, we found that CHAC1 exhibited similar proteinase K sensitivity to those proteins resident within the intermembrane space (IMS) (Fig. 4d). Collectively, these data suggest that a fraction of CHAC1 is within the mitochondria, providing localized GSH catabolism to modulate mitochondrial cysteine availability.

With this confirmation of mitochondrial localization, we next assessed the influence of CHAC1 expression on Fe-S protein function in polyclonal CHAC1 knockout cell lines (Supplementary Fig. 4a)[50]. Loss of CHAC1 protein did not influence ACO2 or ETC complex activities in the presence of cystine (Supplementary Fig. 4b–d). However, CHAC1 loss promoted a significant reduction in the functionality of these Fe-S proteins under cystine deprivation (Supplementary Fig. 4b–d). To supplement these findings, we also employed a doxycycline-inducible shRNA system to knockdown CHAC1 (Supplementary Fig. 5a). Again, we found that CHAC1 loss had no detrimental effect on Fe-S protein function in the presence of cystine (Supplementary Fig. 5b–d). However, unlike in control cells, the activities of ACO2 and the Fe-S dependent respiratory supercomplexes were diminished to varying degrees following 20 h of cystine starvation in CHAC1-deficient cells (Supplementary Fig. 5b–d). Furthermore, a complementary temporal assessment of ACO2 and supercomplex I-III activities revealed that CHAC1 loss accelerated the decay of their function under extended starvation (Fig. 4e, f). Consistently, we also found that CHAC1 loss expedited the reduction in Fe-S protein expression in response to prolonged cystine starvation (Fig. 4g). Importantly, reconstitution of CHAC1-deficient cells with either untargeted or mitochondrially targeted CHAC1 (Fig. 4h, i) rescued the Fe-S protein defects elicited by cystine starvation (Fig. 4j–l).

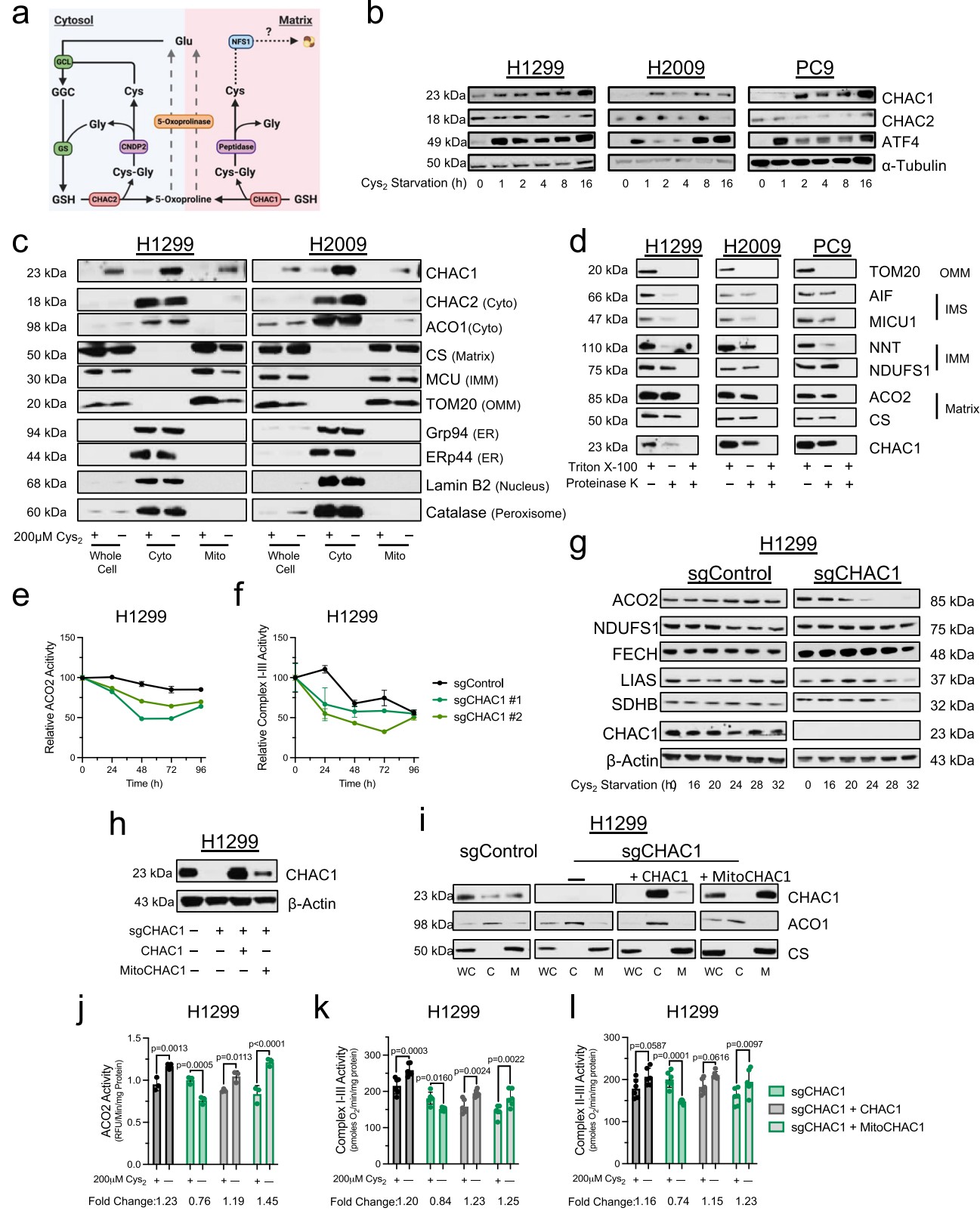

To ensure that these effects were specific to CHAC1, we also evaluated the influence of CHAC2 knockdown on Fe-S protein function (Supplementary Fig. 5e). Critically, CHAC2 loss had no effect on ACO2 or respiratory chain activity regardless of cystine availability (Supplementary Fig. 5f–h). Collectively these data indicate that CHAC1 supports Fe-S protein function under cysteine starvation through an enhanced association with the mitochondria. Further, this activity is confined to CHAC1, as its homolog CHAC2 is absent from the mitochondria.

## CHAC1 supplies the matrix cysteine pool in the absence of extracellular cystine

To determine if CHAC1 has a discernable influence on the pools of mitochondrial cysteine and GSH in the potential support of Fe-S

**Fig. 4 | CHAC1 loss diminishes Fe-S protein function under Cys₂ starvation.**
**a** Schematic of the intracellular GSH cleavage system; Cys, cysteine, Cys₂, cystine, Cys-Gly, cysteinylglycine, GGC, γ-glutamylcysteine, Gly, glycine, Glu, glutamate, GSH, glutathione. **b** Immunoblot analysis of CHAC1, CHAC2, ATF4, and α-Tubulin expression in NSCLC cells under cystine starvation. **c** Immunoblot analysis of CHAC1, CHAC2, ACO1, CS, MCU, TOM20, Grp94, Erp44, Catalase, and Lamin B2 expression in whole cell or fractionated lysates upon mitochondrial immunoprecipitation from cells cultured in the indicated [cystine] for 20 h. **d** Immunoblot analysis of TOM20, AIF, MICU1, NNT, NDUFS1, ACO2, CS, and CHAC1 in mitochondrial isolates subject to proteinase K treatment in the presence or absence of detergent; IMM inner mitochondrial membrane, IMS intermembrane space, OMM outer mitochondrial membrane. **e** ACO2 ($n = 3$ biologically independent samples per condition, except when $n = 2$ for sgControl-96h) and (**f**), supercomplex I–III activities ($n = 3$ biologically independent samples per condition) in CHAC1 expressing or deficient H1299 cells following treatment with cystine-deficient media supplemented with 1 μM ferrostatin-1 for the indicated time. **g** Immunoblot analysis of ACO2, NDUFS1, FECH, LIAS, SDHB, CHAC1, and β-Actin in CHAC1

expressing or deficient H1299 cells following culture in cystine-deficient media supplemented with 1 μM ferrostatin-1 for the indicated time. **h** Immunoblot analysis of CHAC1 and β-Actin expression in control or CHAC1-deficient H1299 cells reconstituted with vector control, untargeted, or mitochondrially-targeted CHAC1. **i** Immunoblot analysis of CHAC1, ACO1 (cytosol), and CS (mitochondria) expression in fractionated lysates from cells in (**h**); C cytosol, M mitochondria, WC whole cell. **j** ACO2 activities ($n = 3$ biologically independent samples per condition) and (**k, l**) ETC supercomplex activities ($n = 6$ biologically independent samples per condition) in control or CHAC1-deficient H1299 cells reconstituted with vector control, untargeted, or mitochondrially-targeted CHAC1 following 20 h culture with or without cystine. For (**e, f**) data points are representative of the activity of each Fe-S protein relative to cystine-replete conditions (time 0 h). Data represent mean values ± s.d. For (**j–l**) data are representative of at least 3 experimental replicates and $P$ values were calculated using a two-way ANOVA. **a** created with BioRender.com released under a Creative Commons Attribution-NonCommercial-NoDerivs 4.0 International license, https://creativecommons.org/licenses/by-nc-nd/4.0/deed.en. For (**b–l**), source data is provided as a Source Data file.

cluster synthesis, we endeavored to monitor the availability of these metabolites in response to cystine starvation (Fig. 5a). To accomplish this, we employed an established liquid chromatography-mass spectroscopy (LC-MS) metabolomics scheme that makes use of HA-Mito cells to facilitate the rapid isolation of mitochondria for analysis of matrix metabolites[47,51]. Though this system enables the detection of the wide range of metabolites present within the mitochondria, the nature of this scheme prevents the detection of cysteine due to the high reactivity of its resident thiol[47,52]. To overcome this technical limitation, we utilized the alkylating agent n-ethylmaleimide (NEM)[53] to stabilize thiol containing metabolites. Application of NEM rapidly alkylated cellular thiols, indicated by the stabilization of reduced PRDX3 within 30 s of treatment (Supplementary Fig. 6a). Importantly, this did not interfere with the capacity to isolate mitochondria from HA-Mito cells (Supplementary Fig. 6b).

We found that this NEM derivatization indeed permitted the detection of mitochondrial cysteine (Fig. 5b). Consistent with our previous findings[5], cystine starvation promoted the rapid depletion of cellular cysteine (Fig. 5b). The rate of exhaustion was similar between the cytosol and matrix, with a half-life of less than 30 min across compartments. In contrast, cellular GSH levels declined more gradually over the course of 20 h, with the matrix pool being marginally more stable than the cytosol following starvation (Fig. 5c). Next, to scrutinize the effects of both CHAC1 and Fe-S cluster synthesis on mitochondrial cysteine availability under limiting conditions, we assessed matrix cysteine in CHAC1 and/or NFS1-deficient H2009-HA-Mito cells subject to a 2 h starvation period (Fig. 5d). We found that loss of NFS1 significantly spared matrix cysteine (Fig. 5e), indicating NFS1 catabolism of cysteine in support of Fe-S cluster synthesis is a robust modulator of the mitochondrial cysteine pool. In contrast, we found that CHAC1 knockdown enhanced the depletion of mitochondrial cysteine under 2 h of cystine starvation (Fig. 5e). This is further evidence to indicate that mitochondrial-associated CHAC1 may reside within the IMS to support matrix uptake of GSH-derived cysteine (Fig. 4d). Moreover, these data reinforce that the expedited loss of Fe-S protein function in the absence of CHAC1 is associated with a deficiency in de novo cluster synthesis.

We then evaluated the effect of CHAC1 knockdown on GSH availability at 12 h of cystine starvation. Both cytosolic and matrix GSH levels were elevated in the absence of CHAC1 (Fig. 5f and Supplementary Fig. 6c, d). Interestingly, NFS1 loss promoted a discernable retention of matrix GSH under 12 h of cystine starvation but did not further enhance the significant effect that CHAC1 had on mitochondrial GSH (Fig. 5f). Unlike what we observed after 2 h starvation, CHAC1 loss no longer had an effect on matrix cysteine availability after 12 h of cystine starvation, yet cytosolic cysteine was modestly spared in CHAC1-deficient cells (Supplementary Fig. 6e, f). Together, these data

indicate that CHAC1 catabolism of GSH has a significant influence on cysteine and GSH availability across compartments under cystine starvation.

## Maintenance of Fe-S clusters is antagonistic to survival under cystine starvation

To determine if the GSH-dependent maintenance of Fe-S protein function plays a contributing role to ferroptosis in NSCLC, we evaluated cell viability under cystine starvation in response to the disruption of Fe-S cluster homeostasis. Beyond synthesis, Fe-S clusters are regulated at the level of their oxidation status (Fig. 6a)[54,55]. We found that knockdown of components of the core Fe-S cluster biosynthetic complex diminished Fe-S protein function and prolonged survival of H1299 and PC9 cells starved of cystine (Fig. 6b and Supplementary Fig. 7a–c). A previous report found that loss of Fe-S cluster synthesis exacerbated erastin-induced ferroptosis in A549 cells due to an accumulation of iron[56]. Consistent with this report, we find that NFS1 knockdown provides no survival benefit under cystine starvation in A549 cells and that ISCU knockdown exacerbates ferroptosis in these cells (Supplementary Fig. 7a, d). This suggests a cell line dependent response to the targeted disruption of the Fe-S cluster synthesis machinery under cystine starvation. We previously established that the mitochondrial NADPH producing nicotinamide nucleotide transhydrogenase (NNT) supports Fe-S protein function through the mitigation of cluster oxidation[40]. We observed that the decrease in respiratory chain function upon NNT knockdown was associated with an increase in survival under cystine deprivation (Fig. 6c and Supplementary Fig. 7e–g). Together these data suggest that the maintenance of Fe-S cluster homeostasis is detrimental to cystine-starved NSCLC cells.

We next evaluated if the defect in Fe-S protein function observed with CHAC1 loss under cystine deprivation was also associated with a delay in ferroptosis. Indeed, CHAC1 knockout significantly prolonged survival under cysteine deprivation (Fig. 6d), and was associated with a decrease in plasma membrane lipid peroxidation (Supplementary Fig. 7h). Importantly, reconstitution of CHAC1-deficient cells with either untargeted or mitochondrially-targeted CHAC1 fully reversed this protective effect (Fig. 6e), indicating that the mitochondrial fraction of CHAC1 is relevant to its pro-ferroptotic activity under cysteine limitation. To extend this analysis, we determined if CHAC1 knockout influenced the response to other ferroptosis inducers. We found that CHAC1 knockout cells were similarly protected against inhibition of cystine uptake with erastin, but not against the GPX4 inhibitor RSL3 (Fig. 6f). This further suggests that the effect of CHAC1 on ferroptosis induction is at least in part due to its effect on the mitochondria, as only cysteine restriction and not GPX4 inhibition-induced ferroptosis has an established mitochondrial component[15]. Finally, we employed

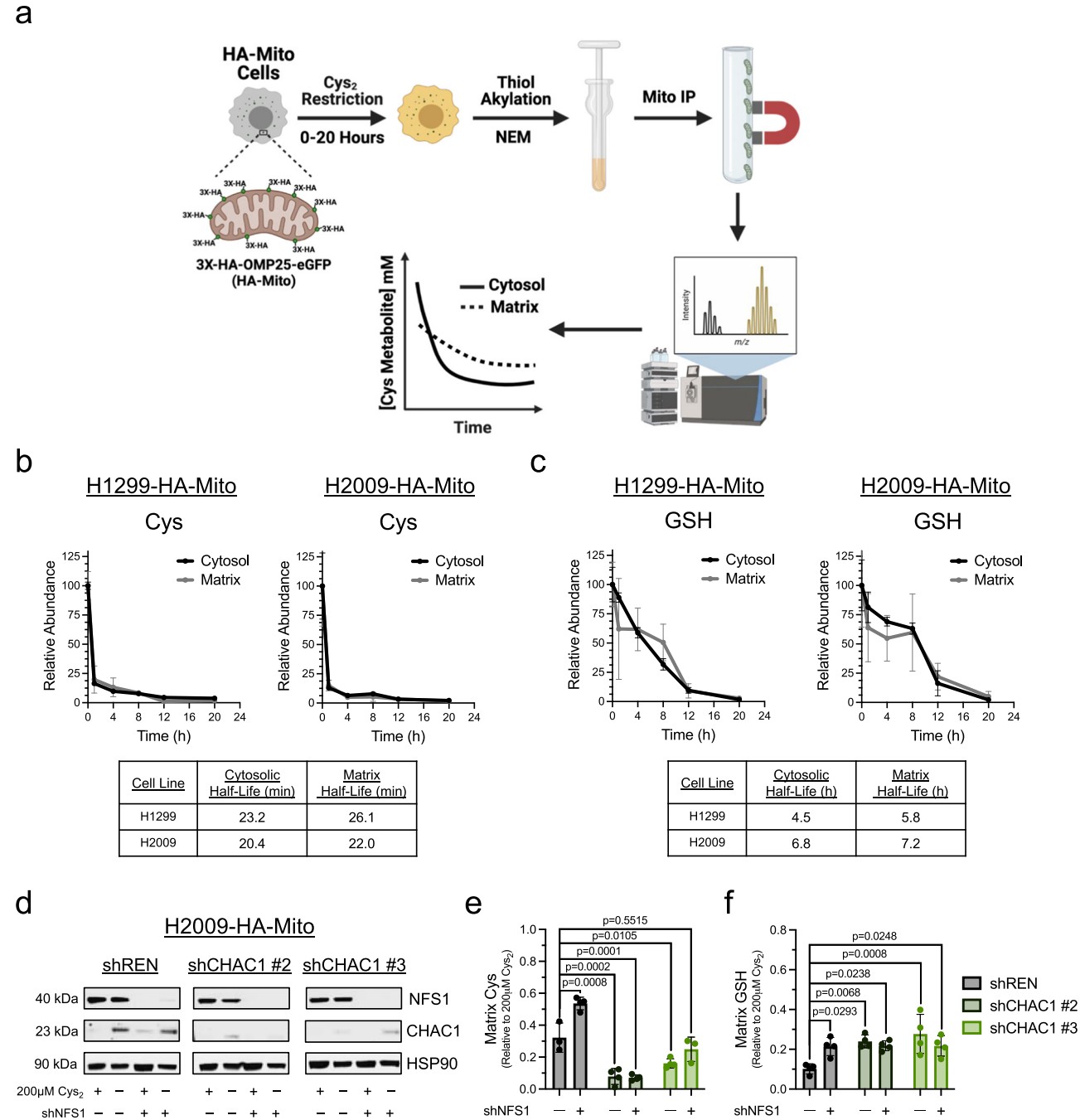

**Fig. 5 | CHAC1 supports the matrix cys under Cys₂ deprivation. a** Schematic workflow of mitochondrial isolation coupled to LC-MS for the compartmentalized detection of thiol-containing metabolites; Mito IP, mitochondrial immunoprecipitation. NEM, n-ethylmaleimide. **b** Determination of the half-life of cytosolic and matrix Cys in H1299-HA-Mito and H2009-HA-Mito cells cultured in the absence of extracellular Cys for 20 h ($n = 3$ biologically independent samples per time point). **c** Determination of the half-life of cytosolic and matrix GSH in H1299-HA-Mito and H2009-HA-Mito cells cultured in the absence of extracellular Cys for 20 h ($n = 3$ biologically independent samples per time point). **d** Immunoblot analysis of NFS1, CHAC1, and β-Actin expression in cystine-fed or starved H2009-HA-Mito cells 3-days post-lentiviral infection with either a scramble control or NFS1-targeting hairpin that were also subject to 5-days of 0.2 μg/mL doxycycline treatment to induce shRNA-mediated knockdown of CHAC1. **e** Matrix cysteine levels in H2009-HA-Mito cells deficient in CHAC1 and or NFS1 that were subjected to 2 h of cystine starvation ($n = 4$ biologically independent samples per condition, except when $n = 3$ for shREN + scramble and shCHAC1 #3 + shNFS1). **f** Matrix GSH levels in H2009-HA-Mito cells deficient in CHAC1 and or NFS1 that were subjected to 12 h of cystine starvation ($n = 4$ biologically independent samples per condition). For (**e**, **f**) data are representative of the matrix pool of each metabolite relative to cystine replete conditions. Data represent mean values ± s.d. For (**e**, **f**) $P$ values were calculated using a two-way ANOVA with a Šidák's multiple comparisons test. **a** created with BioRender.com released under a Creative Commons Attribution-NonCommercial-NoDerivs 4.0 International license, https://creativecommons.org/licenses/by-nc-nd/4.0/deed.en. For (**b**–**f**), source data provided as a Source Data File.

inhibitors of complex I (rotenone) and complex III (myxothiazol) to confirm if the consequence of Fe-S protein function - sustained ETC flux - impacts the induction of ferroptosis. In support of this, we found that both rotenone and myxothiazol prolonged survival in the absence of cystine (Fig. 6g). In totality, our findings indicate that NSCLC cells sustain their respiratory capacity in response to cysteine deprivation to the detriment of their viability. This is achieved through the maintenance of mitochondrial cysteine metabolism in the form of Fe-S

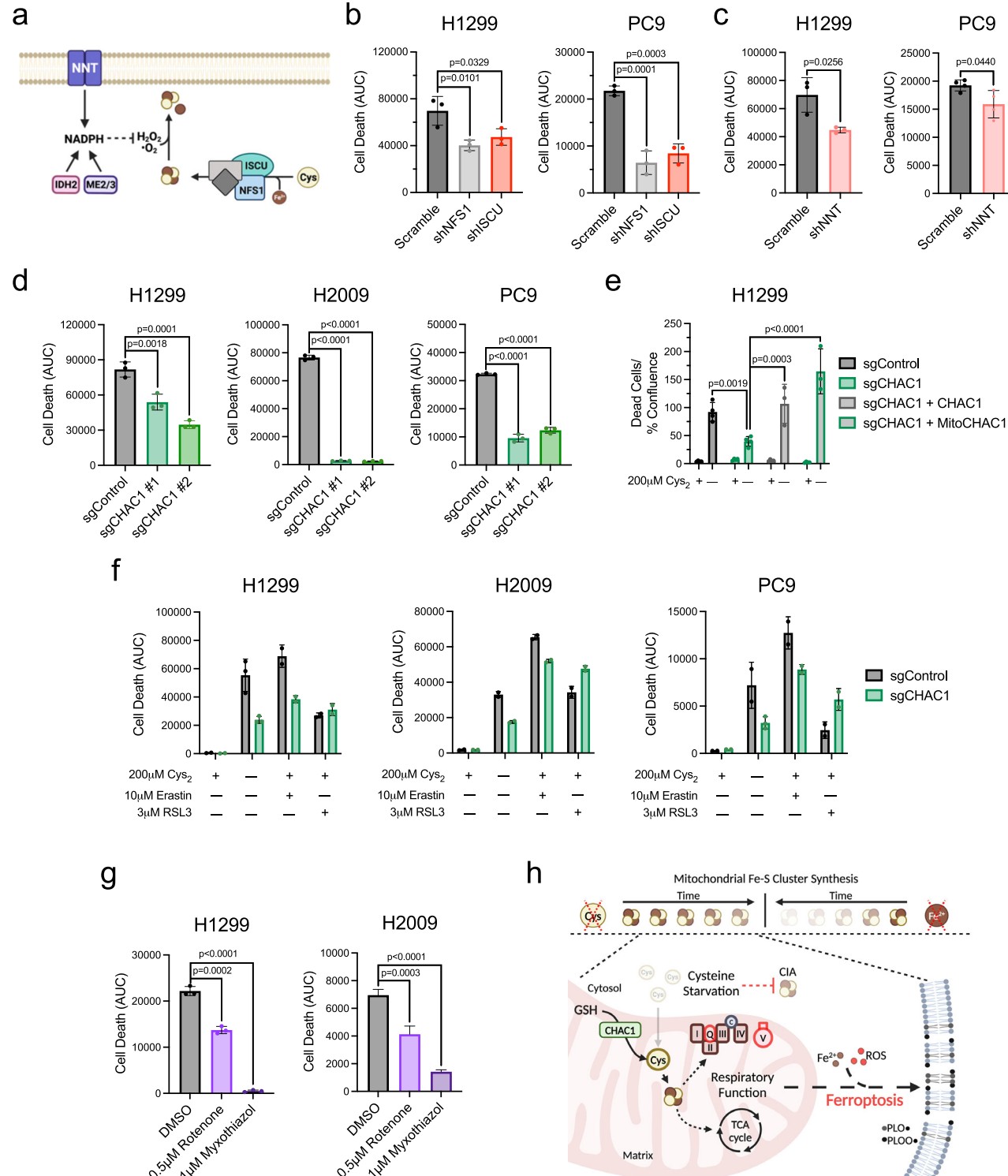

cluster synthesis and depends on the catabolism of GSH by CHAC1 (Fig. 6h).

## Discussion

Mitochondria are critical mediators of the cellular response to various stressors, integrating both intrinsic and extrinsic signals to modulate their function in support of cell resilience[57]. However, there exist instances in which mitochondria compete with the broader cellular milieu to selfishly benefit themselves at the level of the mitochondrial genome[58]. This is likely an ancient relic of endosymbiosis and has a significant impact on human disease[58–60]. Our findings indicate a selfish

response to nutrient stress in NSCLC at the level of mitochondrial metabolism. Herein we describe a self-defeating maintenance of respiratory function through the persistence of mitochondrial cysteine metabolism in the absence of an extracellular source. This occurs through an apparent exploitation of the cellular stress response; where the activation of ATF4 in response to cystine starvation drives the expression of CHAC1[42,44].

CHAC1 is recognized as a potential biomarker for ferroptosis[6,61] and has been implicated in execution of this cell death pathway[61]. However, the mechanism by which CHAC1 promotes ferroptosis has not been adequately defined. Our work demonstrates that CHAC1

**Fig. 6 | Mitochondrial respiratory function antagonizes NSCLC viability in the absence of extracellular Cys2. a** Schematic representation of the two major aspects of Fe-S cluster homeostasis: de novo synthesis and mitigation of cluster oxidation; Cys, cysteine, $Fe^{2+}$, iron, $H_2O_2$, hydrogen peroxide, NADPH, nicotinamide adenine dinucleotide phosphate, $\cdot O_2^-$, superoxide. **b, c** Quantification of cell death over 48 h of cystine starvation in H1299 and PC9 cells following the disruption of Fe-S cluster homeostasis through shRNA knockdown of (**b**) NFS1 and ISCU ($n = 3$ biologically independent samples per condition), or (**c**) NNT ($n = 3$ biologically independent samples per condition for H1299, and $n = 4$ for PC9). **d** Measurement of CHAC1-deficient NSCLC cell death under 48 h of cystine starvation ($n = 3$ biologically independent samples per condition). **e** Quantification of cell death in control, CHAC1-defcient, or CHAC1-reconstituted H1299 cells following a 48 h incubation in the presence or absence of cystine ($n = 4$ biologically independent samples per condition, except when $n = 3$ for sgCHAC1 + CHAC1-0 μM and sgCHAC1 + MitoCHAC1-0 μM). **f** Analysis of CHAC1-deficient NSCLC cell death over 48 h of treatment with indicated inducers of ferroptosis ($n = 2$ biologically independent samples per condition, except when $n = 3$ for H1299 + sgControl-DMSO-0 μM, H1299 + sgCHAC1-DMSO-0 μM, and PC9 + sgCHAC1-DMSO-0 μM). **g** Quantification of cell death over 48 h in H1299 and H2009 cells subject to ETC inhibition under cystine starvation ($n = 3$ biologically independent samples per condition). **h** Summary schematic depicting that mitochondrial Fe-S cluster synthesis is more resistant to sulfur (Cys) than iron restriction due to CHAC1 catabolism of mitochondrial GSH. This contrasts with the cytosolic iron-sulfur cluster assembly system (CIA), which exhibits sensitivity to Cys starvation. The persistence of respiratory function under Cys limitation potentiates the iron-mediated generation of membrane phospholipid alkoxyl (PLO•) and peroxy (PLOO•) radicals that mediate ferroptosis; GSH glutathione, ROS reactive oxygen species. Data represent mean values ± s.d. For (**b–g**) data are representative of at least 3 experimental replicates. For (**b–d**) and (**g**) *P* values were calculated using a one-way ANOVA. For (**e**) *P* values were calculated using a two-way ANOVA with a Šidák's multiple comparisons test. **a, h** created with BioRender.com released under a Creative Commons Attribution-NonCommercial-NoDerivs 4.0 International license, https://creativecommons.org/licenses/by-nc-nd/4.0/deed.en. For **b–g**, source data provided as a Source Data file.

catabolism of GSH is a significant factor in the depletion of GSH across cellular compartments. In the cytosol of NSCLC cells this is accompanied by an accumulation of ROS[5], which facilitates peroxidation of plasma membrane lipids[8]. Intriguingly, we find that the depletion of matrix GSH is not associated with an accumulation of oxidative stress in NSCLC mitochondria. Instead, the regulated degradation of GSH by CHAC1 actually sustains mitochondrial function through the support of Fe-S cluster synthesis in the absence of a soluble pool. However, this persistence of mitochondrial function under cysteine limitation may be a unique characteristic of NSCLC compared to other malignancies. Though mitochondrial metabolism is implicated in ferroptosis, mitochondrial function is reported to progressively decline upon prolonged cysteine deprivation, marked by drastic changes in membrane potential, damage to the IMM, and fragmentation of the mitochondrial network in fibrosarcoma, breast, and liver cancers[15,22,61]. Moreover, the systemic depletion of cysteine promoted severe mitochondrial morphological changes and loss of cristae in pancreatic ductal carcinoma[3]. The propensity for lung cancer cells to retain their mitochondrial function under this nutrient stress may be a factor of their tissue of origin. The lung experiences atmospheric oxygen tension upon inspiration, saturating the tissue with this vital nutrient[62]. However, oxygen is incredibly toxic in excess, and the mitochondria serve a critical function in coupling its detoxification to the generation of energy[63]. Alveolar type II cells, which are appreciated as a cell of origin for lung adenocarcinoma[64,65], are enriched with mitochondria to manage the highly oxygenated lung microenvironment[66]. This may contribute to the substantial mitochondrial metabolism exhibited by human lung tumors[25,26] and the necessity for mitochondrial function in lung tumorigenesis[40,67–70]. Considering the importance of mitochondrial function in lung tumors, our data may reflect an inherent prioritization of mitochondrial function in response to stress in NSCLC. This prioritization of mitochondrial function causes an apparent misalignment with the capacity to maintain cell viability under cysteine starvation, as our findings strongly reinforce the connection between ETC activity and ferroptosis[15–18]. Still, the discrete mitochondrial output that mediates ferroptosis remains unresolved.

Though only a fraction of total CHAC1 is associated with the mitochondria in our cells, we show that mitochondrial CHAC1 supports the matrix cysteine pool and is sufficient to sustain Fe-S protein activity under cystine starvation. In the absence of CHAC1, cysteine insufficiency for Fe-S cluster synthesis occurs more rapidly than under CHAC1-replete conditions, resulting in expedited loss of Fe-S protein function and expression. In light of a recent report uncovering a significant connection between matrix GSH and Fe-S proteins at the organismal level[71], our findings indicate coincident importance of mitochondrial GSH on both sides of the IMM in regulating Fe-S cluster biology under various conditions and agree with an aspect of this

regulation being at the level of cluster synthesis itself[72]. Furthermore, our data reinforce the compartmentalized nature of Fe-S cluster synthesis itself, as analysis of extramitochondrial Fe-S protein stability indicated varied sensitivity towards cysteine limitation that was not seen with mitochondrial proteins. These cytosolic and nuclear Fe-S proteins require a distinct cytosolic iron-sulfur cluster assembly system (CIA)[21] that can involve the maturation of mitochondrially-exported clusters[73] or de novo synthesis within the cytosol[74–76]. Though we did not distinguish the extent to which cysteine restriction influences these two routes of cytosolic cluster synthesis, our findings strongly reflect previous work demonstrating a sensitivity of the CIA to GSH limitation, as GSH is required in the assembly and maturation of cytosolic clusters[77,78]. Still, our data raise the possibility of several additional nodes of regulation of the CIA in response to cysteine limitation that are worthy of further exploration, including inhibition of cytosolic cluster synthesis due to cysteine insufficiency, restriction of mitochondrial cluster export, or differential cluster integration into recipient proteins through the regulation of the chaperone-based trafficking system[79]. Alternatively, the loss of extramitochondrial Fe-S protein stability may be a consequence of Fe-S cluster oxidation due to cytosolic ROS accumulation[56,80].

Our work further implicates Fe-S protein dependent metabolism in the induction of ferroptosis, yet this does not restrict the influence of CHAC1 on other mitochondrial cysteine metabolism that may be relevant to ferroptosis. CHAC1 maintenance of matrix cysteine under limiting conditions could also support the generation of $H_2S$ and subsequent production of hydropersulfides, which potently mitigate lipid peroxidation[81]. Enhanced hydropersulfide production would contribute to the observed lack of mitochondrial oxidative stress under cystine starvation and support the associated maintenance of mitochondrial metabolic function. Regardless, our finding that expression of mitochondrially-targeted CHAC1 is sufficient to restore ferroptosis sensitivity in CHAC1-deficient cells indicates that the maintenance of matrix cysteine availability is operative in the induction of ferroptosis. Still, this does not preclude extramitochondrial CHAC1 from concurrently influencing ferroptosis, as we show that CHAC1 loss significantly spares cytosolic GSH under cystine starvation, a significant factor in the capacity of cells to mitigate lipid peroxidation[11]. Interestingly, we also observe a modest retention of cytosolic cysteine under starved conditions in CHAC1-deficient cells. This may be a consequence of the significant sparing of cytosolic GSH, which lessens the demand for and imposes negative feedback on the de novo synthesis pathway[82]. Although CHAC1 catabolism of cytosolic GSH is detrimental in the context of cysteine limitation, this explains why CHAC1 expression is restricted under basal conditions[42,44,83]. CHAC1 exhibits 20X the catalytic efficiency of CHAC2, which is constitutively expressed for the maintenance of GSH at millimolar levels

within the cell[5,84]. Therefore, CHAC1 likely serves a cytoprotective role when excess GSH is detrimental to the cell, such as conditions of reductive stress[85,86]. Given our findings, a characterization of CHAC1 function in response to various redox stressors is highly warranted.

## Methods

### Cell culture

Human lung adenocarcinoma cell lines (A549, H322, H1299, H1792, H1975, H2009, PC9) were previously obtained[87] from the Harmon Cancer Center Collection (University of Texas-Southwestern Medical Center) or commercially from the American Type Culture Collection (ATCC) and cultured in RPMI 1640 medium (Fisher Scientific, 11-875-119) supplemented with 5% FBS (Sigma-Aldrich, F0926) in the absence of antibiotics. Cell lines were maintained at 37 °C in a humidified incubator containing 95% air and 5% $CO_2$ and routinely tested for mycoplasma contamination with the MycoAlert Assay (Lonza, LT07-418). For experiments requiring manipulation of cystine availability, cells were cultured in RPMI 1640 medium without L-glutamine, L-cysteine, L-cystine, and L-methionine (Fisher Scientific, ICN1646454) supplemented with 5% dialyzed FBS (Sigma-Aldrich, F0392), L-glutamine (VWR, VWRL0131-0100), and L-methionine (Sigma-Aldrich, M5308). L-Cystine (Sigma-Aldrich, C6727) was also added back to the deficient media when required. Additional reagents used for the described experiments included: DMSO (VWR, 97063), FCCP (Sigma-Aldrich, C2920), ferrostatin-1 (Cayman Chemical, 17729), Z-VAD-FMK (Fisher Scientific, F7111), cyclosporin A (Fisher Scientific, NC9676992), deferoxamine (DFO; Sigma-Aldrich, D9533), erastin (Cayman Chemical, 17754), RSL3 (Cayman Chemical, 19288), rotenone (Sigma-Aldrich, R8875), myxothiazol (Neta Scientific, T5580).

### Viral infection

For the generation of lentivirus, Lenti-X 293 T cells (Clontech, 632180) were cultured in DMEM (Fisher Scientific, MT10013CV) supplemented with 5% FBS to 90% confluency. They were then co-transfected in the presence of polyethylenimine (Sigma-Aldrich, 408727) with 6 μg of the plasmid of interest and 6 μg of the packaging plasmids pCMV-VSV-G (Addgene, 8454) and pCMVdR8.2 dvpr (Addgene, 8455) in a 1:8 ratio. Cells were infected in the presence of 8 μg/mL polybrene (Santa Cruz Biotechnology, sc-134220A) with lentivirus for 8 h at an optimized dilution determined with the Lenti-X GoStix Plus protocol (Takara, 631280). Mitochondrially-targeted catalase (MitoCatalase) was cloned into the pLenti-CMV-Blast vector (Addgene, 17445) and introduced into H1299 cells as previously described[40]. Knockout cells were generated using the pLenti-CRISPR-V2 plasmid (Addgene, 52961) encoding validated sgRNA sequences towards CHAC1[88]. Knockout cells were selected with 1 μg/mL puromycin (Invivogen, ant-pr-1) for 5d prior to use for experimentation. Reconstitution of CHAC1 or MitoCHAC1 in CHAC1-knockout cells involved cloning of the respective nucleotide sequences purchased as gBlocks from Integrated DNA Technologies into the pLenti-CMV-GFP-Hygro vector (Addgene, 17446). The Mito-CHAC1 nucleotide sequence was designed such that the initiating codon was replaced with the coding sequence for the first 25 amino acids of the ornithine transcarbamylase leader sequence[89] to facilitate targeting of exogenous CHAC1 to the mitochondrial matrix. Cells were then infected with GFP control, CHAC1, or MitoCHAC1 lentivirus and then selected with 100 μg/mL hygromycin (Invivogen, ant-hg-1) for 5d prior to use for experimentation. Additionally, shRNA sequences targeting CHAC1[90] or CHAC2 were cloned into a modified version of the doxycycline inducible LT3GEPIR vector[50], where the puromycin resistance cassette was replaced with a zeocin resistance gene to generate cell lines with temporal control of CHAC1 depletion. These cells were selected in 100 μg/mL zeocin (Invivogen, ant-zn-1) for 7d prior to use for experimentation. Finally, shRNA sequences targeting NFS1 (Addgene, 102963), ISCU (Addgene, 102972), NNT (Open Biosystems, TRCN0000028507), or a nontargeting control sequence (Scramble;

Millipore Sigma, SHC002) in a pLKO.1 vector were used to disrupt the expression of proteins associated with Fe-S cluster homeostasis. Cells were selected in 1 μg/mL puromycin for 3d and then used for experimental analysis 4d post-infection.

For the generation of 3XHA-EGFP-OMP25 (HA-Mito) retrovirus[47], Phoenix-AMPHO 293 T cells (ATCC, CRL-3213) were cultured in DMEM supplemented with 5% FBS to 90% confluency. They were then transfected with 6 μg of the pMXs-3XHA-EGFP-OMP25 (Addgene, 83356) plasmid in the presence of polyethylenimine. Cells were infected in the presence of 8 μg/mL polybrene with undiluted retrovirus for 24 h, then overlaid with fresh retrovirus for an additional 24 h. Retrovirally-infected cells were then selected with 10 μg/mL blasticidin (Invivogen, ant-bl-1) for 5d prior to use for experimentation.

### Seahorse analyses of mitochondrial function

Analyses of mitochondrial metabolic function were performed with a Seahorse XFe96 Analyzer and Seahorse Wave software (Version 2.4.3; Agilent Technologies, Santa Clara, CA, USA). General mitochondrial function was assessed based on the Seahorse XF Cell Mito Stress Test protocol (Agilent Technologies, Santa Clara, CA, USA). Basal OCR was determined by subtracting rotenone and antimycin A (Sigma-Aldrich, A8674) insensitive oxygen consumption from baseline measurements. Coupling efficiency was determined by calculating the proportion of rotenone/antimycin A sensitive oxygen consumption that is also sensitive to oligomycin (Sigma-Aldrich, 75351) treatment. ATP-linked respiration was determined by calculating the fraction of baseline oxygen consumption that is sensitive to ATP synthase inhibition. Finally, maximal respiratory capacity was determined by subtracting the residual oxygen consumption following rotenone/antimycin A treatment from the oxygen consumption stimulated by mitochondrial uncoupling. Fe-S cluster dependent ETC complex function was assessed according to a previously established protocol[39]. Briefly, 40,000 cells were seeded overnight on quadruplicate wells of an XFe96 microplate. Cells were then washed twice with mitochondrial assay solution (220 mM mannitol [Sigma-Aldrich, M4125], 70 mM sucrose [Sigma-Aldrich, S7903], 10 mM $KH_2PO_4$ [VWR, 470302], 2 mM HEPES [Fisher Scientific, BP310], and 1 mM EGTA [VWR, 97062]) and then overlaid with 175 μL of mitochondrial assay solution supplemented with 10 mM sodium pyruvate (Sigma-Aldrich, P5280), 1 mM malate (Sigma-Aldrich, M0875), 4 mM ADP (Sigma-Aldrich, A5285), and Seahorse Plasma Membrane Permeabilizer (Agilent, 102504-100). Cells were then sequentially subjected to 1 μM rotenone to inhibit complex I, 10 mM succinate (Sigma-Aldrich, S3674) to stimulate complex II, and finally 1 μM antimycin A to inhibit complex III. For assays requiring a period of cystine starvation, 20,000 cells were seeded overnight and then overlaid with cystine replete or deficient RPMI for 20 h prior to starting the assay. Following each assay, cells in each well were lysed to determine protein abundance for normalization.

### Cell viability

Cells were seeded overnight onto black walled 96-well plates at a density of 7500–12,500 cells/well in a total volume of 100 μL. The cells were then washed twice with PBS and then overlaid with cystine-free RPMI supplemented with 20 nM Sytox Green (Fisher Scientific, S7020) and the indicated treatment. Cells were placed in an IncuCyte S3 live-cell analysis system (Essen BioScience, Ann Arbor MI, USA) or a CellCyte X Live Cell Analyzer (Cytena, Freiburg im Breisgau, Germany) housed in a humidified incubator containing 95% air and 5% $CO_2$ at 37 °C. A series of images were acquired every 2–6 h with a 10X objective lens in phase contrast and green fluorescence (Ex/Em: 460/524 nm with an acquisition time of 200-400 ms). Data was processed using IncuCyte S3 (Version 202 A; Essen BioScience, Ann Arbor MI, USA) or CellCyte Studio (Version 5.1; Cytena, Freiburg im Breisgau, Germany) software. Cell death was calculated as the number of Sytox Green positive cells/mm² and normalized to cell confluence (dead cells/mm²/% confluence). Cell

death data were represented as area under the curve (AUC), which represents the sum of dead cells over a 48 h period. When analyzing cell death in cells engineered with green fluorescence, an alternative red fluorescent dye (BOBO-3 iodide; Thermo Fisher Scientific, R37601) was employed. Following the indicated treatment, cells were supplemented with 500 nM BOBO-3 iodide, incubated within the live cell imager for 15 min, and then terminal phase contrast and red fluorescent images (Ex/Em: 580/612 nm with an acquisition time of 1 s) acquired for analysis. Cell death data was calculated as the number of BOBO-3 iodide positive cells per % cell confluence (dead cells/% confluence).

## Fluorescence-based analyses of ROS

Fluorescent probes assessing $\Delta\Psi_m$, mitochondrial density, cellular ROS, mitochondrial $\cdot O_2^-$, $H_2O_2$, and lipid peroxidation were measured by flow cytometry. For each treatment condition, $10^5$ cells were seeded overnight on triplicate wells of a 6-well plate. Cells were then washed twice with PBS and overlaid with cystine replete or deficient RPMI for 20 h. $\Delta\Psi_m$ was determined using tetramethylrhodamine (TMRE; Fisher Scientific, T669) as previously described[91]. At 19.5 h of treatment, cells were incubated in 250 nM TMRE suspended in the appropriate treatment media for 30 min at 37 °C and then collected for analysis. Mitochondrial density was determined using MitoTracker Green FM (ThermoFisher Scientific, M7514) according to the manufacturer's protocol. Following treatment, cells were incubated in 200 nM MitoTracker Green FM in PBS for 15 min at 37 °C and then collected for analysis. Cellular ROS levels were assayed using CellROX Green (ThermoFisher Scientific, C10444) according to the manufacturer's protocol. Briefly, at 19.5 h of treatment, 4 µL of 2.5 mM CellROX Green was applied to each well for a working concentration of 5 µM. Cells were incubated at 37 °C for the remaining 30 min of treatment and then collected for analysis. Mitochondrial $\cdot O_2^-$ was determined with MitoSOX Red (ThermoFisher Scientific, M36008) based on a previously described protocol[92]. Following treatment, cells were incubated in 5 µM of MitoSOX Red in PBS for 20 min at 37 °C and then collected for analysis. Mitochondrial $H_2O_2$ was assessed with MitoPY1 (Fisher Scientific, 442810) according to an established protocol[93]. Following treatment, cells were incubated in 10 µM of MitoPY1 in PBS for 20 min at 37 °C and then collected for analysis. Mitochondrial lipid peroxidation was determined using the ratiometric fluorescent probe MitoPerOX (Cayman Chemical, 18798) according to the manufacturer's protocol. At 19.5 h of treatment, cells were incubated in 10 µM MitoPerOX suspended in the appropriate treatment media for 30 min at 37 °C and then collected for analysis. For all analyses, cells were washed twice with ice cold PBS following incubation with the respective probe and collected into 500 µL of ice cold PBS for analysis with a BD Accuri C6 Plus Flow Cytometer and BD CSampler Plus software (Version 1.0.34.1; BD Biosciences, Franklin Lakes, NJ, USA). For TMRE, and MitoSOX Red, a phycoerythrin (PE) optical filter was used for fluorescence detection. For MitoTracker Green, CellROX Green, and MitoPY1, a fluorescein isothiocyanate (FITC) optical filter was used for fluorescence detection. MitoPerOX fluorescence was detected with both the PE and FITC filters to calculate the ratio of oxidized (green) and unaltered (red) membrane lipids. For each analysis, the mean fluorescence intensity of 10,000 discrete events was calculated (Supplementary Fig. 8). Whole cell lipid peroxidation was determined using the Image-IT Lipid Peroxidation Kit (ThermoFisher Scientific, C10445) according to the manufacturer's protocol. For each treatment condition, 15,000 cells were seeded overnight onto triplicate wells of a black walled 96-well plate. Cells were then washed with PBS and treated with cystine replete or deficient RPMI for 20 h. At 19.5 h of treatment, 2 µL of 10 mM BODIPY-C11 was applied to each well for a working concentration of 10 µM. Cells were incubated at 37 °C for the remaining 30 min of treatment. Cells were washed twice with PBS and then placed in a fluorescence-compatible plate reader and fluorescence measured (reduced Ex:475 nm/Em:580–640 nm, oxidized Ex:475 nm/Em:500–550 nm).

## Mitochondrial GSH:GSSG ratio

The oxidation state of the mitochondrial GSH pool was assessed using a previously developed genetically encoded fluorescent and ratiometric biosensor targeted to the mitochondria[35] (Mito-Grx-roGFP2). The Mito-Grx1-roGFP2 construct (Addgene, 64977) was cloned into the pLenti-CMV-Puro vector (Addgene, 17448) and lentivirus generated to produce stable cell lines expressing the biosensor as described. For each treatment condition, 15,000 cells were seeded overnight onto triplicate wells of a black walled 96-well plate. Cells were then washed with PBS and treated with cystine replete or deficient RPMI for 16 h. Cells were then placed in a fluorescence-compatible plate reader (Promega, Madison, WI, USA) and fluorescence measured (GSH-bound Ex:475 nm/Em:500–550 nm, GSSG-bound Ex:405 nm/Em:500–550 nm).

## Immunoblotting

Following the indicated treatment, cell lysates were generated in ice cold RIPA lysis buffer (20 mM Tris-HCl (VWR, 97061-258), pH 7.5, 150 mM NaCl (Fisher Scientific, S271), 1 mM EGTA, 1 mM EDTA [Sigma-Aldrich, E5134], 1% sodium deoxycholate [Sigma-Aldrich, D6750], 1% NP-40 [Sigma-Aldrich, 74385]) supplemented with protease inhibitors (Fisher Scientific, PIA32955). Protein concentrations were then determined by DC Protein Assay (Bio-Rad, 5000112) and 10−30 µg samples were prepared with a 6x reducing sample buffer containing β-mercaptoethanol (VWR, M131). Proteins were resolved by SDS-PAGE using NuPAGE 4–12% Bis-Tris precast gels (Fisher Scientific, WG1402BOX) and then transferred to 0.45 µM nitrocellulose membranes (VWR, 10120-006). Membranes were then blocked with 5% nonfat dairy milk in Tris-buffered saline containing 0.1% Tween 20 (VWR, 0777), and incubated overnight with the following primary antibodies: ACC (1:1000; Cell Signaling Technologies, 3662, RRID:AB_2219400), ACO1 (1:1000, GeneTex, GTX128976, RRID:AB_2885857), ACO2 (1:1000; GeneTex, GTX109736, RRID:AB_1939567), AIF (1:1000; Cell Signaling Technology, 4642S, RRID:AB_2224542), ATF4 (1:1000; Cell Signaling Technologies, 11815, RRID:AB_2616025), β-Actin (1:2000; Thermo Fisher Scientific, AM4302, RRID:AB_2536382), CS (1:1000; Cell Signaling Technology, 14309S, RRID:AB_2665545), Catalase (1:1000, Cell Signaling Technology, 12980, RRID:AB_2798079), CHAC1 (1:1000; Proteintech, 15207-1-AP, RRID:AB_2878118), CHAC2 (1:1000; GeneTex, GTX128819, RRID:AB_2885821), DLST (1:1000; Cell Signaling Technologies, 11954, RRID:AB_2732907), DPYD (1:1000; Cell Signaling Technology, 4654S, RRID:AB_10614011), ERp44 (1:1000; Cell Signaling Technologies, 3798, RRID:AB_1642195), FECH (1:1000; Santa Cruz Biotechnology, sc-377377, RRID: n/a), Grp94 (1:1000, Cell Signaling Technology, 2104, RRID:823506), HSP60 (1:1000; Cell Signaling Technologies, 4870, RRID:AB_2295614), HSP90 (1:1000; Cell Signaling Technologies, 4874, RRID:AB_2121214), ISCU (1:1000; Santa Cruz Biotechnology, sc-373694, RRID:AB_10918261), Lamin B2 (1:1000, Cell Signaling Technology, 12255, RRID:AB_2797859), LIAS (1:1000; Proteintech, 11577-1-AP, RRID:AB_2135972), Lipoic Acid (1:1000; Millipore Sigma, 437695-100UL, RRID:AB_10683357), MCU (1:1000; Cell Signaling Technologies, 14997, RRID:AB_2721812), MICU1 (1:1000; Cell Signaling Technology, 12524S, RRID:AB_2797943), NDUFS1 (1:1000; Cell Signaling Technologies, 70264, RRID: N/A), NFS1 (1:1000; Santa Cruz Biotechnology, sc-365308, RRID:AB_10843245), NNT (1:1000; Abcam, ab110352, RRID:AB_10887748), NTHL1 (1:1000; Proteintech, 14918-1-AP, RRID:AB_2154555), PDH-E2 (1:1000; Abcam, ab126224, RRID:AB_11129511), POLD1 (1:1000; Abcam, AB186407, RRID:AB_2921290), PPAT (1:1000; Proteintech, 15401-1-AP, RRID:AB_2166532), PRDX3 (1:1000; Abcam, ab73349, RRID:AB_1860862), SDHB (1:1000; Cell Signaling Technologies, 92649, RRID: N/A), SHMT2 (1:1000; Cell Signaling Technology, 12762S, RRID:AB_2798018), TOM20 (1:1000; Santa Cruz Biotechnology, sc-17764, RRID:AB_628381), α-Tubulin (1:1000; Abcam, ab7291, RRID:AB_2241126). HRP-conjugated anti-mouse (1:10,000; Jackson ImmunoResearch, 115-005-003, RRID:AB 2338447) and anti-rabbit

(1:10,000; Jackson ImmunoResearch, 111-005-003, RRID:AB 2337913) secondary antibodies and enhanced chemiluminescence were then used for all immunoblotting.

For the determination of the PRDX3 oxidation state, a previously established redox immunoblotting protocol was employed[94]. Briefly, following the indicated treatment, cells were washed twice with PBS and overlaid with 200 μL of alkylation buffer (40 mM HEPES, 50 mM NaCl, 1 mM EGTA, protease inhibitors) supplemented with 25 mM N-ethylmaleimide (NEM; Fisher Scientific, AA4052606). Cells were incubated for 10 min at room temperature and then 20 μL of 10% CHAPS detergent (Sigma-Aldrich, C3023) was added, and cells incubated at room temperature for an additional 10 min to lyse cells. Lysates were then cleared by centrifugation at 4 °C for 15 min at 17,000 g. Supernatants were isolated to quantify protein and 5–10 μg protein samples were mixed with a 4X non-reducing buffer prior to separation by SDS-PAGE as described.

### ACO activity

Cytosolic (ACO1) and mitochondrial aconitase (ACO2) activities were determined as we have previously described[40]. Briefly, cystine fed or starved cells were collected and resuspended in 200 μL of 50 mM Tris-HCl and 150 mM NaCl, pH 7.4. Cells were then homogenized by dounce homogenizer (15 strokes) and the homogenate spun down at 4 °C for 10 min at 10,000 g. The supernatant (cytosolic fraction) was then collected and the organellar pellet washed twice and resuspended in 100 μL of 1% Triton X-100 (VWR, 0694) in 50 mM Tris-HCl and 150 mM NaCl, pH 7.4 to lyse the mitochondrial membrane. Lysates were then spun down at 4 °C for 15 min at 17,000 g. Protein concentrations were then determined by DC Protein Assay, and 175 μL of 100–500 μg/mL protein were generated in 50 mM Tris-HCl, pH 7.4 and incubated at 37 °C for 15 min. 50 μL of this protein solution was transferred to triplicate wells of a black walled 96-well plate containing 50 μL of 50 mM Tris-HCl, pH 7.4. Next, 50 μL of a 4 mM NADP+ (Neta Scientific, SPCM-N1131-OM) and 20 U/mL IDH1 (Sigma-Aldrich, I1877) solution, and 50 μL of 10 mM sodium citrate were sequentially added to initiate the assay. The plate was transferred to a fluorescence-compatible plate reader maintained at 37 °C to measure the change in NADPH auto-fluorescence every minute over a period of 45 min. The accumulation of NADPH autofluorescence over time is an indicator of aconitase activity, where ACO1/2 present in the sample converts the supplied citrate to isocitrate, which is subsequently metabolized by the supplied IDH1 in a reaction that yields NADPH.

### Mitochondrial immunoprecipitation

An established protocol for the rapid isolation of mitochondria from HA-Mito cells[47] was modified as follows to preserve thiol containing metabolites and permit their detection by LC-MS. Prior to mitochondrial isolation, 50-200 μL of Pierce Anti-HA magnetic beads (VWR, PI88837) were washed in KPBS (136 mM KCl [Sigma-Aldrich, P5405], 10 mM KH₂PO₄, pH 7.4 in HPLC grade H₂O [Fisher Chemical, W5-1]) for each sample. Samples were processed one at a time in a 4 °C cold room to best preserve the metabolic state of each sample. Cells were washed twice with ice cold KPBS and then incubated in KPBS supplemented with 25 mM NEM and 10 mM ammonium formate (Fisher Scientific, 501454965) for 1 min. Cells were then washed twice more and collected into ice cold KPBS. Cells were pelleted at 1,000 g for 2 min and then resuspended in 200 μL of KPBS. Cells were homogenized by Dounce homogenizer (25 strokes), and the homogenate spun down at 1000 g for 2 min. The supernatant was then applied to the pre-washed magnetic beads and incubated on an end-over-end rotator for 3.5 min. Beads were then isolated from solution with a DynaMag-2 magnetic stand (ThermoFisher Scientific, 12321D). The unbound solution was kept as a mitochondrial-free cytosolic fraction. Mitochondria-bound beads were washed twice with KPBS and then processed for either immunoblotting or metabolite detection. For the purposes of

immunoblotting, isolated beads were reconstituted in 50 μL of lysis buffer (50 mM Tris-HCl, pH 7.4, 150 mM NaCl, 1 mM EDTA, 1% Triton X-100, protease inhibitors), incubated on ice for 10 min and lysates cleared by centrifugation at 17,000 g for 10 min. Protein was quantified and samples prepared for SDS-PAGE. For metabolite detection, beads were reconstituted in 50 μL of 80% methanol containing 20 μM of [¹³C₃, ¹⁵N]-NEM-Cysteine (Cambridge Isotope Laboratories, CNLM-38710H-0.25) and 40 μM of [¹³C₂, ¹⁵N]-NEM-GSH (Sigma-Aldrich, 683620), incubated on ice for 5 min, and extracts cleared by centrifugation at 17,000 g for 10 min. NEM-derivatized standards were prepared as previously described[5]. Alternatively, the mitochondrial-free cytosolic fractions were extracted in 800 μL of the defined extraction solution, incubated on ice for 5 min, and cleared by centrifugation at 17,000 g for 10 min.

### Immunofluorescence

To assess the distribution of CHAC1 protein and its colocalization with the mitochondrial compartment, 250,000 cells expressing Mito-Grx1-roGFP2[35] were seeded overnight onto 22 mm circular glass coverslips. Following the indicated treatment, cells on coverslips were gently washed twice with PBS and incubated in 500 μL of 4% paraformaldehyde in PBS (Thermo Scientific, J19943-K2) for 10 min. Fixed cells were then washed three times with PBS and then permeabilized in 500 μL 0.1% Triton X-100 in PBS for 10 min. Permeabilized cells were then washed three times with PBS and blocked with 500 μL of 2% bovine serum albumin (BSA; Fisher Scientific, BP9703-100) for 1 h at room temperature. Blocked cells on coverslips were then incubated overnight in a CHAC1 primary antibody (1:500 in 3% BSA/PBS; Invitrogen, PA5-103719, RRID:AB_2853052) at 4 °C. Coverslips were then washed three times with PBS and incubated in an Alexa Fluor 647-conjugated secondary antibody (1:500 in 3% BSA/PBS; Thermo Fisher Scientific, A31573, RRID:AB_2536183) for 1 h at room temperature in the dark. Coverslips were then washed three times with PBS in the dark and then incubated in 500 μL of 300 nM DAPI (Sigma-Aldrich, D9542) for 5 min at room temperature in the dark. Fully stained cells were then washed three times with PBS in the dark, mounted onto glass coverslips (Fisher Scientific, 125442) with ProLong Gold antifade mounting media (Thermo Fisher Scientific, P10144), and dried overnight in the dark. Cells were then visualized by confocal microscopy with a Leica SP8 Confocal Microscope and Leica Application Suite X software (Version 5.1; Leica Microsystems, Deerfield, IL) equipped with a 63X oil immersion lens. 405 nm, 488 nm, and 638 nm lasers were sequentially employed to excite DAPI, Mito-Grx1-roGFP2, and Alexa Fluor 647 fluorescence, respectively. DAPI and Mito-Grx1-roGFP2 fluorescence were collected with separate PMT detectors, and Alexa Fluor 647 fluorescence was detected with a HyD detector. Mean fluorescence intensity for discrete cells were analyzed with publicly available Fiji software[95]. CHAC1 colocalization with mitochondria was determined with The Coloc 2 Fiji plug-in to determine the correlative overlap (Pearson's Correlation between red (CHAC1) and green (Mito-Grx1-roGFP2) signals) as an indication of CHAC1 colocalization with mitochondria.

### Proteinase K assay

To determine the sub-organellar localization of CHAC1 within the mitochondria, a previously established protocol employing the indiscriminate protease, proteinase K, was followed[49]. Briefly, >20,000,000 cells were harvested in cold PBS and pelleted at 500 g for 3 min at 4 °C. The supernatant was removed, and cells resuspended in 1 mL of Homogenate buffer (350 mM Tris-HCl, pH 7.8, 250 mM NaCl, 50 mM MgCl₂) supplemented with 1 mM phenylmethylsulfonyl fluoride [PMSF; EMD Millipore, 52332]. Cells were then homogenized by Dounce homogenizer (>15 strokes), supplemented with 100 μL of Equilibration buffer (3.5 mM Tris-HCl, pH 7.8, 2.5 mM NaCl, 500 μM MgCl2), and centrifuged at 1200 g for 3 min at 4 °C. Supernatant

containing mitochondria was then transferred to a fresh 1.5 mL microcentrifuge tube and spun down at 15,000 $g$ for 2 min at 4 °C. This supernatant was discarded and the pellet containing mitochondria was resuspended in 250 μL of buffer A (10 mM Tris-HCl, pH 7.4, 1 mM EDTA, 320 mM Sucrose). Mitochondria were then isolated by centrifugation at 15,000 $g$ for 2 min at 4 °C. The mitochondrial pellet was resuspended in 90 μL of 1X TD (49.99 mM Tris base [Research Products International, T60040-5000], 274.13 mM NaCl, 20.12 mM KCl, 13.95 mM $Na_2HPO_4$ [Sigma-Aldrich, S5136]) and split between three 1.5 microcentrifuge tubes. The first tube was then supplemented with 3.33 μL of 20% Triton X-100, the second with 3.33 μL of 250 μg/mL proteinase K (Thermo Fisher Scientific, EO0491), and the third tube with 3.33 μL of both 20% Triton X-100 and 250 μg/mL proteinase K. Samples were then incubated at room temperature for 20 min with mixing by pipette every 5 min. All samples were then supplemented with 1 mM PMSF to inactivate proteinase K, volumes normalized with 1X TD, and prepared for SDS-PAGE.

### LC-MS-based metabolite detection

For determination of the relative Cys and GSH pool sizes across compartments, previously established LC-MS conditions were employed[5,24] For chromatographic metabolite separation, a Vanquish UPLC system was coupled to a Q Exactive HF (QE-HF) mass spectrometer equipped with HESI (Thermo Fisher Scientific, Waltham, MA, USA). Samples were run on an Atlantis Premier BEH Z-HILIC VanGuard FIT column, 2.5 μm, 2.1 mm × 150 mm (Waters, Milford, MA, USA). The mobile phase A was 10 mM $(NH_4)_2CO_3$ and 0.05% $NH_4OH$ in $H_2O$, while mobile phase B was 100% ACN. The column chamber temperature was set to 30 °C. The mobile phase condition was set according to the following gradient: 0-13 min: 80% to 20% of mobile phase B, 13–15 min: 20% of mobile phase B. The ESI ionization mode was negative, and the MS scan range (m/z) was set to 65-975. The mass resolution was 120,000 and the AGC target was $3 \times 10^6$. The capillary voltage and capillary temperature were set to 3.5 kV and 320 °C, respectively. 5 μL of sample was loaded. For each experimental run, 5 μL of extraction solvent were run in triplicate as blank controls; these were disbursed throughout the run to signal changes in the integrity of metabolite detection. The LC-MS metabolite peaks were manually identified and integrated by El-Maven (Version 0.10.0) by matching with a previously established in-house library[5].

### Statistical analysis

All data were analyzed for statistical significance with GraphPad Prism 9.4.1 software; $P < 0.05$ were deemed significant (n.s., not significant, *, $P < 0.05$, **, $P < 0.01$, ***, $P < 0.001$, ****, $P < 0.0001$). Comparisons of two groups were performed with a two-tailed unpaired Student's $t$-test. For comparisons of a single variable in 3 or more groups a one-way ANOVA with a Dunnet's multiple comparisons test was performed. For comparisons of two variables a two-way ANOVA with a Sidak's multiple comparisons test was performed.

### Reporting summary

Further information on research design is available in the Nature Portfolio Reporting Summary linked to this article.

## Data availability

All data supporting the findings of this study are provided within the figures and Source Data files. Source data are provided with this paper.

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

## Acknowledgements

We would like to thank Laura Torrente for cloning the zeomycin-resistant version of the LT3GEPIR vector for this study and members of the DeNicola and Ana Gomes laboratories for their very helpful discussions. Further, we thank the laboratories of John Cleveland, Elsa Flores, Bob Gillies, Ana Gomes, and Vince Luca for access to their equipment in support out this study. This work was supported by grants from the NIH/NCI (R37CA230042 and P01CA250984) to G.M.D. This work was also supported by the Analytical Microscopy and Proteomics/Metabolomics Cores at H. Lee Moffitt Cancer Center & Research Institute, which are funded in part by Moffitt's Cancer Center Support Grant (NCI, P30-CA076292). All schematics were created with BioRender.com.

## Author contributions

N.P.W. conceived the study, designed and performed the experiments, analyzed the data, and wrote the manuscript; S.J.Y. performed LC-MS analysis of thiol metabolites; T.F. generated CHAC1-knockout lines and performed immunoblotting; A.M.S. characterized CHAC1 localization in response to $Cys_2$ limitation and performed immunoblotting; M.A.O. cloned sgCHAC1-resistant CHAC1 and MitoCHAC1 constructs, generated and characterized (Mito)CHAC1-reconstituted cell lines, and performed immunoblotting; J.M. characterized CHAC1 localization and expression in response to $Cys_2$ starvation; G.M.D. conceived the study, acquired funding, contributed to the experimental design, supervised the project, and wrote the manuscript.

## Competing interests

The authors declare no competing interests.
