## [Peer Review File · Nature Communications]

Mitochondrial Respiratory Function is Preserved Under Cysteine Starvation via Glutathione Catabolism in NSCLCEditorial Note: This manuscript has been previously reviewed at another journal that is not operating a transparent peer review scheme. This document only contains reviewer comments and rebuttal letters for versions considered at *Nature Communications*.

REVIEWER COMMENTS

Reviewer #1 (Remarks to the Author):

-Western blot in 4j is not high quality enough to assess the impact of CHAC1 knockout. SDHB levels seem quite similar, indicating that sgCHAC1 may not have any impact on iron sulfur clusters. There is clearly a technical immunoblotting issue here. These experiments should at least be repeated several times or quantitated.

- The authors did not perform necessary sgRNA rescue experiments. This is typically a golden standard in biomedical papers, but should be an editorial decision.

Reviewer #2 (Remarks to the Author):

The authors have thoughtfully addressed all of my comments. I congratulate them on their important discoveries and I support acceptance / publication at this time.

Reviewer #3 (Remarks to the Author):

The authors have done admirable work to address reviewers' comments on their previous submission. While many questions are satisfactorily answered. Several issues remain:

1. Although new IF and blot data lend stronger argument that there is a small portion of CHAC1 localized in mitochondria, whether this is the portion of CHAC1 responsible for the proposed function remains to be determined. Is it possible to test this hypothesis by engineering a CHAC1-KO cell line and then reconstituting the KO cells with artificial CHAC1 that is localized strictly in mitochondria or cytoplasm?
2. The authors argued (reasonably) that due to rapid use of cysteine in mitochondria, cystine deprivation depletes both cytosolic and mitochondrial cysteine equally rapidly. However, since the central hypothesis of the study deals with the prolonged availability of mitochondrial cysteine for Fe-S cluster synthesis, they need to perform additional experiments to measure mitochondrial cysteine to compare the following conditions, with or without cystine deprivation from culture medium: -/+ CHAC1 KO -/+ blockage of Fe-S synthesis. There should be a clear difference distinguishing these four conditions, if their main hypothesis is correct.
3. Overall, the major experimental results are consistent with their mitochondria CHAC1-mitochondrial cysteine-Fe/S cluster hypothesis. However, while a functional and causative relationship from cystine

deprivation all the way to mitochondrial cysteine level is testable (providing the authors satisfy points 1 and 2), whether the final functional outcome (mitochondrial function and cell death response) is mainly due to Fe-S cluster proteins is purely correlative. And unfortunately, one does not see an obvious/practical way to experimentally establish the causative relationship here. As such, one way to deal with it is for the authors to carefully discuss this issue; this reviewer will not hold the lack of almost impossible causative experiments against the authors, but definitely would like to see a more detailed discussion and a somewhat tuned-down conclusion.

Reviewer #4 (Remarks to the Author):

The manuscript addresses an interesting aspect of sulfur metabolism in human cells (here mainly NSCLC cells). It is claimed that depletion of Cys does not immediately lead to a dysfunction of Cys-dependent processes in mitochondria (like mitochondrial translation (not tested though) or FeS biosynthesis by the ISC proteins) because the huge GSH pool is used for generating additional Cys via CHAC1 dependent degradation. This sounds to be a really attractive model, but in general the effects are rather weak throughout the manuscript and the data scatter in individual experiments is often so high that I am not convinced of this interpretation yet (see below for an example). ISC defects usually generate strong defects in FeS proteins and this is not seen here which may mean that the right experimental conditions have not been found. My feeling is that the authors used a way too short Cys depletion time to analyze their samples). Generally, what I also miss in this study is a validation of the many tools/assays that are used in the study. For example, if "mitochondrial superoxide" is measured, I would like to see what effect might be expected if there is indeed oxidative stress in mitochondria (i.e. positive and negative controls). How do we know that all the assays work reliably and what maximum signal the assays would generate. Since the effects are usually very small, this is a major drawback of the entire study, and makes me worry how reliable the individual parts of the measurements are. The authors did not look at extra-mitochondrial FeS proteins. Numerous groups have shown that mitochondria are essential for maturation of these factors and the effects are often more pronounced than those inside mitochondria. The authors should analyze a few cytosolic and/or nuclear FeS proteins (easy assays are available, like Aco1 measurement or blotting). As mentioned below, the localization of CHAC1 to mitochondria in starved cells needs more rigorous controls to be convincing. In the following I have listed some points that came up during reading of the manuscript.

1. Line 115/6: „... suggesting cysteine is acting as a respiratory substrate“. This statement is confusing, since the compound at best is used indirectly for respiration. This statement needs semantic improvement.
2. I suppose that the experiments of Figs. 1 and 2 are done under 20 h of Cys depletion. Is this enough time for depletion because ExData Fig. 1 says this is just the start of the phenotype development. I would therefore think the experiments need to be redone for a longer (30 or 40 h) depletion. This applies to many experiments.
3. Why the A549 cells are always outliers in these assays, remains really obscure. Maybe these cells are simply more sensitive than other cells and need less time to develop the phenotypes? Also for this

question, a more time-resolved study seems to be appropriate. A generalization of the manuscript's message is therefore not possible.

4. The statement in line 123/4 saying "persistence of respiratory function is not damaging to the mitochondria of NSCLC cells" is confusing. If a function is maintained, why should this be damaging. The issue needs a better rationale (e.g., higher membrane potential, ...).

5. The description of the depletion experiments in Fig. 3f-g plus ExData Fig. 2a are really confusing. Do I get it right that the time 0 in Fig. 3f-g corresponds to 48 h of NFS1-ISCO depletion? Then, the ISC depletion effects are really weak, in comparison to published data, eg., from the Rouault and Lill labs. Anyway, I would suggest to describe the experiments only on a single time-scale without a resetting to zero.

6. Why did the authors fail to measure Aco activity during the depletion time course for the ISC components?

7. The correct localization of CHAC1 in mitochondria is unclear for a number of reasons. The blot of Fig. 4c localizing CHAC1 to mitochondria is not conclusive. A control for soluble matrix proteins (e.g., ACO2, ISC proteins) is missing (do the authors analyze intact or broken mitos from which matrix has leaked out? This often happens). More importantly, CHAC1 could simply stick to the outside of mitochondria. Therefore, to verify the statement that "CHAC1 is expressed within mitochondria" (line 242), a treatment of the isolated HA-mitos with proteinase K (+/- detergent) is needed to check CHAC1's potential protease protection and its internal vs. external localization. Determination of the sub-mitochondrial localization of CHAC1 seems important, because the authors do not provide any mechanistic explanation for the relocalization of CHAC1 INTO mitos under Cys starvation conditions. There is no recognizable mito targeting sequence within the N-terminus of this protein, and proteins do not simply move into another compartment. The protein could just stick to the outside of the mitochondrial outer membrane. Nevertheless, to explain the observed (weak) effects of CHAC1 on mitochondrial FeS biosynthesis, an attachment of CHAC1 to the outer surface of mitochondria might be sufficient, providing GSH-released Cys to the mito import system. However, in the present text of the manuscript, this idea is conveyed differently. This aspect is the more important as the fluorescence data are not convincing to me (I was part of several stories correcting the localization of ER and mito proteins, and fluorescence data were not reliable enough to conclude on the actual localization).

8. Lanes 267-9: Wrong Callouts. Also later for ExData Fig. 5.

9. I am a bit skeptical about the results shown in ExData Fig. 4. The differences are really small (usually 10% or less). Is this biologically meaningful? For FeS biogenesis clearly not (see above). More importantly, the differences of wild-type and the two KD experiments sometimes seems similarly high as the differences +/- Cys. This all may have to do with the fact that the authors used a wrong time window for Cys depletion (way too short).

10. The severe drawback of the experiments shown in Fig. 5d-g is the extreme data scatter; see for instance 5e and g where the matrix Cys GSH levels of individual data points differ by up to a factor of 4. The authors admit this by stating "considerable intragroup variability was observed in matrix samples due to the inherent technical challenge of assaying thiols". In such cases, only better data acquisition and not statistics does help. I am hesitant to draw any conclusions from such experiments.

11. As shown in Fig. 5c, there is hardly any GSH left over in the cell after 20h. I have a hard time to believe that this low level will cause only the meager effects on FeS biosynthesis (also protein translation should be affected).

12. The experiments in Fig. 5i-k are not convincingly showing that restricting GSH import into

mitochondria (low GSH levels are not directly confirmed; and there is a second transporter!) inhibits FeS biosynthesis. The effects are either very weak (less than 10% drop) or even absent (5i right).

Reviewer #1 (Remarks to the Author):

-Western blot in 4j is not high quality enough to assess the impact of CHAC1 knockout. SDHB levels seem quite similar, indicating that sgCHAC1 may not have any impact on iron sulfur clusters. There is clearly a technical immunoblotting issue here. These experiments should at least be repeated several times or quantitated.

As suggested by reviewer #1, we have repeated this experiment and generated higher quality immunoblots. Our new data now conclusively demonstrates that CHAC1 knockout enhances Fe-S protein loss under cystine starvation (**Figure 4g**). In addition to NDUFS1 and SDHB, we have now analyzed additional Fe-S proteins, including ACO2, LIAS and FECH. Our new data demonstrate that the stability of these proteins is generally more sensitive to cysteine restriction in the absence of CHAC1.

- The authors did not perform necessary sgRNA rescue experiments. This is typically a golden standard in biomedical papers, but should be an editorial decision.

In response to this request by Reviewer #1, and also the request by Reviewer #3 to target CHAC1 to the mitochondria, we have now performed the suggested rescue experiments by reconstituting CHAC1-knockout cells with either untargeted or mitochondrially targeted CHAC1 (**Figures 4h and i**). We find that both versions of CHAC1 rescue the mitochondrial Fe-S protein defects observed under cystine starvation (**Figures 4j-i**). Collectively with our orthogonal shRNA data, this demonstrates that CHAC1 indeed modulates mitochondrial Fe-S protein function under cysteine starvation and bolsters our overall conclusion that CHAC1 catabolism of GSH supplies mitochondrial cysteine in the absence of an extracellular source. Additionally, reconstitution with either version of CHAC1 reversed the protective effect of CHAC1 loss on cell viability under cystine starvation (**Figure 6e**), indicating that the mitochondrial fraction of CHAC1 is relevant to its pro-ferroptotic activity under cysteine limitation.

Reconstitution of CHAC1-Deficient Cells with CHAC1 Rescues Fe-S Protein Defects Precipitated by Cystine Starvation. (a) CHAC1 expression in CHAC1-deficient H1299 cells that were infected with lentivirus encoding GFP control, untargeted CHAC1, or mitochondrially-targeted CHAC1. (b) Subcellular localization of CHAC1 in CHAC1-reconstituted cells. (c) ACO2 activities in control, CHAC1-deficient, and CHAC1-reconstituted H1299 cells cultured for 20h in Cys₂ replete or starved conditions. (d) Supercomplex I-III activities in control, CHAC1-deficient, and CHAC1-reconstituted H1299 cells cultured for 20h in Cys₂ replete or starved conditions. (e) Supercomplex II-III activities in control, CHAC1-deficient, and CHAC1-reconstituted H1299 cells cultured for 20h in Cys₂ replete or starved conditions. (f) Analysis of cell death in control, CHAC1-deficient, and CHAC1-reconstituted H1299 cells following a 36h incubation in Cys₂ replete or starved conditions.

Reviewer #2 (Remarks to the Author):

The authors have thoughtfully addressed all of my comments. I congratulate them on their important discoveries and I support acceptance / publication at this time.

We greatly appreciate the reviewer for reviewing our revised manuscript, and are pleased to hear that we have sufficiently addressed their critiques.

Reviewer #3 (Remarks to the Author):

The authors have done admirable work to address reviewers' comments on their previous submission. While many questions are satisfactorily answered. Several issues remain:

We thank the reviewer for their continued efforts in providing helpful critiques of our manuscript. We have performed the suggested experiments and feel that the new data addressing the reviewer's concerns has greatly strengthened the manuscript. Please find below specific responses to each of the critiques with references to where the corresponding edits appear in the revised manuscript.

1. Although new IF and blot data lend stronger argument that there is a small portion of CHAC1 localized in mitochondria, whether this is the portion of CHAC1 responsible for the proposed function remains to be determined. Is it possible to test this hypothesis by engineering a CHAC1-KO cell line and then reconstituting the KO cells with artificial CHAC1 that is localized strictly in mitochondria or cytoplasm?

We have now performed the suggested rescue experiments by reconstituting CHAC1-knockout cells with either untargeted or mitochondrially targeted CHAC1 (**Figures 4h and i**). We find that expression of both versions of CHAC1 rescue the Fe-S protein defects induced by 20h cystine starvation (**Figures 4j-l**). The observation that restricting CHAC1 to the mitochondria is sufficient to sustain Fe-S protein function under cystine starvation strengthens our claim that CHAC1 supports Fe-S protein function under cysteine limitation through the regulation of mitochondrial cysteine availability. Furthermore, we find that reconstituting CHAC1-knockout cells with mitochondrially-targeted CHAC1 is sufficient to fully reverse the protective effect of CHAC1 loss on viability under cystine starvation (**Figure 6e**), supporting our conclusion that the retention of Fe-S cluster synthesis and mitochondrial function contributes to ferroptosis in NSCLC.

Reconstitution of CHAC1-Deficient Cells with CHAC1 Rescues Fe-S Protein Defects Precipitated by Cystine Starvation. (a) CHAC1 expression in CHAC1-deficient H1299 cells that were infected with lentivirus encoding GFP control, untargeted CHAC1, or mitochondrially-targeted CHAC1. (b) Subcellular localization of CHAC1 in CHAC1-reconstituted cells. (c) ACO2 activities in control, CHAC1-deficient, and CHAC1-reconstituted H1299 cells cultured for 20h in Cys₂ replete or starved conditions. (d) Supercomplex I-III activities in control, CHAC1-deficient, and CHAC1-reconstituted H1299 cells cultured for 20h in Cys₂ replete or starved conditions. (e) Supercomplex II-III activities in control, CHAC1-deficient, and CHAC1-reconstituted H1299 cells cultured for 20h in Cys₂ replete or starved conditions. (f) Analysis of cell death in control, CHAC1-deficient, and CHAC1-reconstituted H1299 cells following a 36h incubation in Cys₂ replete or starved conditions.

2. The authors argued (reasonably) that due to rapid use of cysteine in mitochondria, cystine deprivation depletes both cytosolic and mitochondrial cysteine equally rapidly. However, since the central hypothesis of the study deals with the prolonged availability of mitochondrial cysteine for Fe-S cluster synthesis, they need to perform additional experiments to measure mitochondrial cysteine to compare the following conditions, with or without cystine deprivation from culture medium: +/- CHAC1 KO +/- blockage of Fe-S synthesis. There should be a clear difference distinguishing these four conditions, if their main hypothesis is correct.

We have now performed the suggested metabolomics experiment examining matrix cysteine availability in response to 2h cystine starvation with respect to the presence or absence of CHAC1/NFS1 (Figures 5d-f). We find that NFS1 knockdown spares matrix cysteine over this 2h starvation period, consistent with NFS1 catabolism of cysteine for Fe-S cluster synthesis being a modulator of the matrix pool. In contrast, we observe that CHAC1 loss exacerbates the depletion of matrix cysteine under starvation. This aligns with our new data indicating that mitochondrial CHAC1 is resident within the IMS (Figure 4d), where its catabolism of GSH would provide a localized pool of cysteine for uptake into the matrix. Thus, in the absence of CHAC1, the restriction of matrix cysteine under cystine starvation is amplified by impeding the mobilization of cysteine stored within GSH.

CHAC1 Loss Hastens Depletion of Matrix Cysteine Under Cystine Starvation. (a) Immunoblot analysis of CHAC1 and NFS1 expression in Cys₂-fed or starved H2009-HA-Mito cells subject to NFS1 and/or CHAC1 knockdown. **(b)** Relative matrix Cys levels following 2h Cys₂ starvation in H2009-HA-Mito cells subject to NFS1 and/or CHAC1 knockdown.

3. Overall, the major experimental results are consistent with their mitochondria CHAC1-mitochondrial cysteine-Fe/S cluster hypothesis. However, while a functional and causative relationship from cystine deprivation all the way to mitochondrial cysteine level is testable (providing the authors satisfy points 1 and 2), whether the final functional outcome (mitochondrial function and cell death response) is mainly due to Fe-S cluster proteins is purely correlative. And unfortunately, one does not see an obvious/practical way to experimentally establish the causative relationship here. As such, one way to deal with it is for the authors to carefully discuss this issue; this reviewer will not hold the lack of almost impossible causative experiments against the authors, but definitely would like to see a more detailed discussion and a somewhat tuned-down conclusion.

We contend that the totality of our data is supportive of the conclusion that maintenance of mitochondrial Fe-S proteins under cysteine deprivation contributes to ferroptosis. We have refrained from claiming that persistent mitochondrial function is the only factor in the induction of ferroptosis in NSCLC, as we recognize the multifactorial nature of this form of cell death (**Lines 45-58**). However, our findings align with the accumulating evidence of a mitochondrial link to ferroptosis (PMID: 26166707; 30581146; 32029897; 33493440). We argue that our data showing that the direct inhibition of Fe-S cluster synthesis or mitochondrial electron transport prolong survival under cystine starvation is causative evidence to indicate that mitochondrial metabolic function is a relevant factor in the induction of ferroptosis (**Figures 6b and g**). Furthermore, our observation that expression of mitochondrially-targeted CHAC1 is sufficient to fully reverse the anti-ferroptotic effect of losing the endogenous protein greatly supports our claim for a connection between sustained Fe-S cluster synthesis and ferroptosis.

Still, we recognize the possibility that the maintenance of Fe-S cluster synthesis is coincident to other cysteine metabolism within the mitochondria that may be relevant to the induction of ferroptosis. For instance, CHAC1 maintenance of matrix cysteine could also support H₂S production and the subsequent generation of hydropersulfides, which mitigate lipid peroxidation (PMID: 36109647). This would promote the sustained mitochondrial function and lack of oxidative stress that we observe under cystine starvation (**Figures 1 and 2**). We have added additional discussion to acknowledge this possibility (**Lines 448-454**).

Reviewer #4 (Remarks to the Author):

The manuscript addresses an interesting aspect of sulfur metabolism in human cells (here mainly NSCLC cells). It is claimed that depletion of Cys does not immediately lead to a dysfunction of Cys-dependent processes in mitochondria (like mitochondrial translation (not tested though) or FeS biosynthesis by the

ISC proteins) because the huge GSH pool is used for generating additional Cys via CHAC1 dependent degradation. This sounds to be a really attractive model, but in general the effects are rather weak throughout the manuscript and the data scatter in individual experiments is often so high that I am not convinced of this interpretation yet (see below for an example). ISC defects usually generate strong defects in FeS proteins and this is not seen here which may mean that the right experimental conditions have not been found. My feeling is that the authors used a way too short Cys depletion time to analyze their samples). Generally, what I also miss in this study is a validation of the many tools/assays that are used in the study. For example, if “mitochondrial superoxide” is measured, I would like to see what effect might be expected if there is indeed oxidative stress in mitochondria (i.e. positive and negative controls). How do we know that all the assays work reliably and what maximum signal the assays would generate. Since the effects are usually very small, this is a major drawback of the entire study, and makes me worry how reliable the individual parts of the measurements are. The authors did not look at extra-mitochondrial FeS proteins. Numerous groups have shown that mitochondria are essential for maturation of these factors and the effects are often more pronounced than those inside mitochondria. The authors should analyze a few cytosolic and/or nuclear FeS proteins (easy assays are available, like Aco1 measurement or blotting). As mentioned below, the localization of CHAC1 to mitochondria in starved cells needs more rigorous controls to be convincing. In the following I have listed some points that came up during reading of the manuscript.

We thank the reviewer for their considerable effort in reviewing our manuscript. We have performed additional experiments to address their critiques, which we feel have greatly strengthened the manuscript. This new data indicates that mitochondrial CHAC1 associates with the intermembrane space (IMS), where catabolism of GSH supports matrix uptake of cysteine under cystine starvation. Please find below specific responses to each of the critiques with references to where the corresponding edits appear in the revised manuscript.

i. Generally, what I also miss in this study is a validation of the many tools/assays that are used in the study. For example, if “mitochondrial superoxide” is measured, I would like to see what effect might be expected if there is indeed oxidative stress in mitochondria (i.e. positive and negative controls). How do we know that all the assays work reliably and what maximum signal the assays would generate.

We have an established publication record demonstrating our capacity for assessing oxidative stress in the context of lung cancer (PMID: 21734707; 32007910; 32196080; 33357455; 35667246), and validation of these assays are found within many of these studies. For the current study, these controls were not included in figures for simplicity but we have now included them in the supplementary data, including assay-relevant oxidants or following genetic manipulation of cellular antioxidant capacity (**Extended Data Figures 1f-j**). We are confident that we have properly optimized the assays and tools that we have chosen to implement for this study and contend that the data we have generated indicate that there is indeed no oxidative stress within the mitochondrial compartment within the designated time frame of our analyses.

Validation and optimization of Fluorescent ROS Probes. (a) Relative CellROX Green fluorescence in NSCLC cells treated with PBS or 25µM tert-butyl hydroperoxide (tbHP) for 30 minutes. (b) Relative MitoSOX Red fluorescence in H1299 cells treated with DMSO or 25µM menadione for 30 minutes. (c) Relative MitoPY1 fluorescence in NSCLC cells treated with PBS or 20µM H₂O₂ for 30 minutes. (d) Relative MitoSOX Red fluorescence in control or SOD2-overexpressing A549 cells. (e) Relative MitoPY1 fluorescence in vector-control or MitoCatalase infected H1299 cells challenged with 25µM H₂O₂.

ii. The authors did not look at extra-mitochondrial FeS proteins. Numerous groups have shown that mitochondria are essential for maturation of these factors and the effects are often more pronounced than those inside mitochondria. The authors should analyze a few cytosolic and/or nuclear FeS proteins (easy assays are available, like Aco1 measurement or blotting).

We have now included additional immunoblot data evaluating extramitochondrial Fe-S proteins in response to cystine starvation (**Extended Data Figure 2c**). Similar to the mitochondrial Fe-S proteins we assessed (**Figure 3k**), we find that the stability of these cytosolic and nuclear Fe-S proteins is generally retained through 20h of cystine starvation, further supporting our conclusion that Fe-S cluster synthesis is sustained under the period of cystine starvation used for our study.

Extramitochondrial Fe-S Protein Stability is Sustained Through 20h of Cys₂ Starvation. Time course analysis of cytosolic and nuclear Fe-S protein expression in response to Cys₂ starvation H1299 and H2009 cells.

1. Line 115/6: „... suggesting cysteine is acting as a respiratory substrate“. This statement is confusing, since the compound at best is used indirectly for respiration. This statement needs semantic improvement.

We have revised the manuscript text to clarify that cysteine catabolism can support respiration through the generation of both pyruvate and H₂S (**Lines 117-120**).

2. I suppose that the experiments of Figs. 1 and 2 are done under 20 h of Cys depletion. Is this enough time for depletion because ExData Fig. 1 says this is just the start of the phenotype development. I would therefore think the experiments need to be redone for a longer (30 or 40 h) depletion. This applies to many experiments.

As the reviewer indicates, the onset of our phenotype (i.e., ferroptosis) occurs just beyond 20h of cystine starvation. Given that this phenotype is cell death, we are restricted from extending the duration of our treatment time such not to introduce the substantial confounding influence of an activated cell death pathway on our analyses. Extending the starvation time beyond 20h would require supplementation with ferrostatin-1. While this strategy permitted the evaluation of Fe-S protein function through a prolonged starvation period of up to 96h (**Figures 4e-f**), the antioxidant nature of this anti-ferroptotic agent precludes any study of oxidative stress under prolonged starvation, as the results would be uninterpretable. Figures 1 and 2 clearly reflect that mitochondrial function persists at least through the induction of ferroptosis and that this occurs independent of mitochondrial oxidative stress.

3. Why the A549 cells are always outliers in these assays, remains really obscure. Maybe these cells are simply more sensitive than other cells and need less time to develop the phenotypes? Also for this question, a more time-resolved study seems to be appropriate. A generalization of the manuscript's message is therefore not possible.

We have previously published an extensive characterization of the response of a large panel of NSCLC cell lines to cystine starvation and find that A549 cells are not more sensitive than the average NSCLC cell line (PMID: 33357455), which we have also recapitulated here (**Extended Data Figure 1a**). We stand by our contention that the outlier status of a single line among a broader panel reflects the inherent heterogeneity of human cancer cell lines and does not negate the conclusion that in general NSCLC cells exhibit a persistence of mitochondrial function in the absence of extracellular cysteine. Though they are an outlier regarding respiratory function, we show that Fe-S protein function is maintained under starvation in a manner consistent with the broader panel (**Figures 3b-e**). We have now included additional data demonstrating the association between CHAC1 and Fe-S protein function under cystine starvation in A549 cells as well (**Extended Data Figure 4**), further suggesting that their divergent respiratory response to cysteine limitation is not linked to diminished Fe-S protein function.

CHAC1 Supports Fe-S Protein Function in Cystine Starved A549 Cells. (a) Immunoblot analysis of CHAC1 expression in A549 cells subject to CRISPR/Cas9 knockout of CHAC1. (b) ACO2 activities in CHAC1-expressing or deficient A549 cells subject to Cys₂-replete or starved condition for 20h. (c)

Respiratory supercomplex I-III activities in CHAC1-expressing or deficient A549 cells subject to Cys₂-replete or starved condition for 20h. **(d)** Respiratory supercomplex II-III activities in CHAC1-expressing or deficient A549 cells subject to Cys₂-replete or starved condition for 20h.

4. The statement in line 123/4 saying “persistence of respiratory function is not damaging to the mitochondria of NSCLC cells” is confusing. If a function is maintained, why should this be damaging. The issue needs a better rationale (e.g., higher membrane potential, ...).

We have revised the statement to read “to confirm that the persistence of respiratory function in NSCLC was not associated with mitochondrial stress as has been previously reported in other contexts” (**Lines 125-126**).

5. The description of the depletion experiments in Fig. 3f-g plus ExData Fig. 2a are really confusing. Do I get it right that the time 0 in Fig. 3f-g corresponds to 48 h of NFS1-ISCU depletion? Then, the ISC depletion effects are really weak, in comparison to published data, eg., from the Rouault and Lill labs. Anyway, I would suggest to describe the experiments only on a single time-scale without a resetting to zero.

We contend that visualizing the data for each insult on their own staggered timeline would be even more confusing to readers. Therefore, to resolve this confusion, we have included a schematic representation (**Figure 3f**) of the experimental design for panels 3g-i. shNFS1/ISCU lentiviral infection is now denoted as time point -48h and the first point of analysis for each condition as time point 0h.

Regarding the concern of our results assessing the influence of NFS1/ISCU knockdown on Fe-S protein function, the studies they cite from the Rouault and Lill labs indeed show a “loss” of protein activity upon manipulations of these proteins that may appear more significant than what we have presented here. However, these studies rely on gel-based activity assays to assess the functions of the Fe-S proteins in question (e.g., ACO₂, respiratory complexes) (PMID: 29523684). These assays are not quantitative and have a much lower dynamic range of activity leading to the appearance of a total loss of protein function when that is not necessarily the case. We have employed the Seahorse extracellular flux analyzer to assess the oxygen consumed in association with the specific activities of Fe-S cluster dependent respiratory complexes, providing a quantitative measure of Fe-S protein function. The results we report in **Figures 3g-i** are consistent with what we have previously demonstrated (PMID: 32196080) and of similar magnitude to that reported by the Lill lab when they employed commercial enzyme activity assays (PMID: 34824239). Furthermore, we are restricted in when we can assay these NSCLC cells deficient in Fe-S cluster biosynthesis as they begin to die within five-days post introduction of the shRNA hairpins targeting these proteins. It is likely that this coincides with the near complete loss of Fe-S protein function as cited by the reviewer, however we cannot reliably determine this as our assays require live cells.

6. Why did the authors fail to measure Aco activity during the depletion time course for the ISC components?

We have now included a temporal analysis of ACO₂ activity in response to cystine starvation, DFO treatment, and NFS1/ISCU knockdown as we had done for the Fe-S cluster dependent respiratory complexes (**Figure 3g**).

7. The correct localization of CHAC1 in mitochondria is unclear for a number of reasons. The blot of Fig. 4c localizing CHAC1 to mitochondria is not conclusive. A control for soluble matrix proteins (e.g., ACO₂, ISC proteins) is missing (do the authors analyze intact or broken mitos from which matrix has leaked out? This often happens). More importantly, CHAC1 could simply stick to the outside of mitochondria. Therefore, to verify the statement that “CHAC1 is expressed within mitochondria” (line 242), a treatment of the isolated HA-mitos with proteinase K (+/- detergent) is needed to check CHAC1’s potential protease protection and its internal vs. external localization. Determination of the sub-mitochondrial localization of CHAC1 seems important, because the authors do not provide any

mechanistic explanation for the relocalization of CHAC1 INTO mitos under Cys starvation conditions. There is no recognizable mito targeting sequence within the N-terminus of this protein, and proteins do not simply move into another compartment. The protein could just stick to the outside of the mitochondrial outer membrane. Nevertheless, to explain the observed (weak) effects of CHAC1 on mitochondrial FeS biosynthesis, an attachment of CHAC1 to the outer surface of mitochondria might be sufficient, providing GSH-released Cys to the mito import system. However, in the present text of the manuscript, this idea is conveyed differently. This aspect is the more important as the fluorescence data are not convincing to me (I was part of several stories correcting the localization of ER and mito proteins, and fluorescence data were not reliable enough to conclude on the actual localozaion).

We thank the reviewer for these suggestions, which have provided greater mechanistic insight into CHAC1 mitochondrial localization. We have now repeated our mitochondrial IP experiment and included markers for the outer mitochondrial membrane, inner mitochondrial membrane, and mitochondrial matrix (**Figure 4c**). Importantly, we observe an absence of each mitochondrial protein in the cytosolic fraction, suggesting that we are isolating intact mitochondria for analysis. To resolve the sub-mitochondrial localization, we performed the suggested proteinase K experiment. We find that in the absence of detergent, CHAC1 exhibits protease sensitivity similar to proteins resident within the IMS (**Figure 4d**). Coupled with our metabolomics data showing that CHAC1 loss expedites the depletion of matrix cysteine under cystine starvation (**Figure 5e**), these results indicate that mitochondrial CHAC1 catabolizes GSH within the IMS to support matrix uptake under cystine starvation.

Mitochondrial CHAC1 Localizes to the IMS. (a) Immunoblot analysis of CHAC1 and representative organellar markers in whole cell or fractionated lysates following mitochondrial immunoprecipitation. (b) Immunoblot analysis of mitochondrial isolates from NSCLC cells incubated with proteinase K in the presence or absence of detergent. (c) Immunoblot analysis of CHAC1 and NFS1 expression in Cys₂-fed or starved H2009-HA-Mito cells subject to NFS1 and/or CHAC1 knockdown. (d) Relative matrix Cys levels following 2h Cys₂ starvation in H2009-HA-Mito cells subject to NFS1 and/or CHAC1 knockdown.

8. Lanes 267-9: Wrong Callouts. Also later for ExData Fig. 5.

Thank you for pointing out this error, we have made sure that all figure references are correct in the revised manuscript.

9. I am a bit skeptical about the results shown in ExData Fig. 4. The differences are really small (usually 10% or less). Is this biologically meaningful? For FeS biogenesis clearly not (see above). More importantly, the differences of wild-type and the two KD experiments sometimes seems similarly high as the differences +/- Cys. This all may have to do with the fact that the authors used a wrong time window for Cys depletion (way too short).

The purpose of our study was to demonstrate that CHAC1 influences mitochondrial Fe-S protein function through its catabolism of GSH. Therefore, our analyses of various Fe-S proteins were conducted at biologically relevant time points to prevent the confounding effects of death on our data rather than to identify the maximal effect that the loss of GSH catabolism can exert on these proteins. We contend that while the magnitude of effect for individual groups in some assays is modest, the totality of the Fe-S protein data across multiple cell lines and methods of CHAC1 manipulation demonstrate a consistently measurable effect of CHAC1 loss on these proteins under 20h of cystine starvation. Further, we demonstrate that this effect becomes more pronounced with time when cells are subject to prolonged starvation in the presence of ferrostatin-1 (**Figures 4e and f**). To prevent any misinterpretation of the magnitude of effect observed in our analyses, we have added the fold change to each appropriate panel. This should make it clear that CHAC1 loss elicited a >10% reduction in Fe-S protein function under cystine starvation in the majority of experiments (**19 of 21 total panels across Figures 4j-l and Extended Data Figures 4 and 5**).

10. The severe drawback of the experiments shown in Fig. 5d-g is the extreme data scatter; see for instance 5e and g where the matrix Cys GSH levels of individual data points differ by up to a factor of 4. The authors admit this by stating “considerable intragroup variability was observed in matrix samples due to the inherent technical challenge of assaying thiols”. In such cases, only better data acquisition and not statistics does help. I am hesitant to draw any conclusions from such experiments.

We have performed a new series of metabolomics experiments evaluating the effect of NFS1 and/or CHAC1 knockdown on matrix cysteine and GSH levels under cysteine starvation. The intragroup variability for these experiments was considerably reduced and we observe clear and significant changes in mitochondrial cysteine and GSH levels in CHAC1-deficient cells under cysteine starvation (**Figures 5e and f**).

CHAC1 Loss Modulates Matrix Cysteine and GSH Availability Under Cystine Starvation. (a) Immunoblot analysis of CHAC1 and NFS1 expression in Cys₂-fed or starved H2009-HA-Mito cells subject to NFS1 and/or CHAC1 knockdown. (b) Relative matrix Cys levels following 2h Cys₂ starvation in H2009-HA-Mito cells subject to NFS1 and/or CHAC1 knockdown. (c) Relative matrix GSH levels following 12h Cys₂ starvation in H2009-HA-Mito cells subject to NFS1 and/or CHAC1 knockdown.

11. As shown in Fig. 5c, there is hardly any GSH left over in the cell after 20h. I have a hard time to believe that this low level will cause only the meager effects on FeS biosynthesis (also protein translation should be affected).

While this may be surprising, our data shown in **Extended Data Figure 2d** and **Figures 4e and f** demonstrating that Fe-S protein function in NSCLC cells under prolonged starvation (>24h) suggests that the effect of GSH loss on Fe-S cluster synthesis manifests beyond 20h of starvation. This indicates that a relatively small fraction of the GSH pool can minimally sustain Fe-S cluster synthesis to maintain homeostatic mitochondrial metabolism. We do observe that 20h of cystine starvation is associated with a modest decrease in protein synthesis in H1299 cells, but not H2009 cells.

Protein Synthesis Under Cystine Starvation: Click-IT HPG analysis of protein synthesis in H1299 and H2009 cells subject to 20h Cys₂ starvation.

12. The experiments in Fig. 5i-k are not convincingly showing that restricting GSH import into mitochondria (low GSH levels are not directly confirmed; and there is a second transporter!) inhibits FeS biosynthesis. The effects are either very weak (less than 10% drop) or even absent (5i right).

In line with our other data demonstrating regulation of Fe-S cluster protein function under various conditions (**Extended Data Figures 4, 5, and 7**), we find that SLC25A39 loss reduces Fe-S protein function by >24% across all experiments except ACO2 activity in H2009 cells. However, in light of our new data indicating that mitochondrial CHAC1 localizes to the IMS (**Figures 4d and 5e**), we have removed the SLC25A39 data from the manuscript to simplify the interpretation of our study. The mechanism by which SLC25A39 loss influences Fe-S function is likely distinct from that we observe with CHAC1, and related to the vital role for GSH in Fe-S cluster trafficking and maturation via glutaredoxin 5 (PMID: 16110529). Thus, mitochondrial GSH may play a role in sustaining Fe-S cluster biology on both sides of the inner mitochondrial membrane. Furthermore, it was recently identified that SLC25A39 is subject to Fe-S cluster-dependent autoregulation in the maintenance of both mitochondrial GSH and iron homeostasis that has profound impacts on iron Fe-S cluster biology (PMID: 37917749).

Diminishing Mitochondrial GSH Sensitizes Fe-S Protein Function to Cysteine Restriction. (a) Immunoblot analysis of SLC25A39 deficient H1299 and H2009 cells. (b) Relative matrix cysteine levels in SLC25A39 deficient H1299 cells. (c) ACO2 activities in SLC25A39 expressing or deficient NSCLC cells subject to treatment with Cys₂-replete or deficient media for 20h. (d and e) ETC supercomplex activities in permeabilized SLC25A39-expressing or deficient cells stimulated with d, 10mM pyruvate and 1mM malate or e, 10mM succinate following culture in the presence of absence of cysteine for 20h.

REVIEWER COMMENTS

Reviewer #3 (Remarks to the Author):

The authors have adequately addressed all concerns from this reviewer.

Reviewer #4 (Remarks to the Author):

I read the new manuscript and the rebuttal letter, particularly wrt my concerns. The authors have replied to all of my concerns, and in many cases (see below), the issues are resolved. In a few cases, the issues are still open, but the final experimental addressing may be part of another work. For some cases, however, the authors interpret their (new) data differently from what I see from the presented data. This is a major remaining concern that, in my view, needs to be addressed. For bringing clarity into the entire story, I suggest that the authors draw a model for their views that consistently covers all the observations presented in the manuscript. I am convinced that this would help the authors (and the reviewer as well as the later readers) to judge whether all parts of the findings fit into that model. My specific comments on the rebuttal letter and the revised version of this manuscript are given below (I refer to the sub-points used by the authors in the rebuttal letter):

- i) I appreciate the inclusion of the controls for these ROS probe experiments. This allows the reader to better evaluate the quality of the effects. Resolved.
- ii) The authors immunodetected several cytoplasmic and nuclear FeS proteins. As I was suspecting, many of these FeS proteins showed a marked decline in stability (amount) over the time course. Particularly, DPYD and NTHL (I look at the major band) levels go down, while the data for other proteins are not clear without quantitation of band intensities (and the repeats). I do not understand why the authors interpret this data set as “Fe-S cluster synthesis is sustained under the period of cystine starvation”. The opposite is what I see. This mis-interpretation of the data is more than confusing (mis-leading the reader). Further, I would expect that this FeS protein decline would be reflected in a decrease in enzyme activities, eg cytosolic ACO1. This data is not provided, and is needed here. Hence, my criticism is not resolved, because the interpretation of the story is incorrect, in my view. What the new result, however, means is that GSH depletion (due to its degradation upon Cys starvation) leads to cytosolic FeS defects. This exactly mimics what was seen earlier in yeast (PMID 12011041, 21478822). The title of the manuscript is therefore no longer correct, because biogenesis of cytoplasmic FeS proteins is a key function of mitochondria.

Further note: The headline of ExFig. 2 does not really express what the Figure shows. “Disruption of Fe-S Cluster Synthesis Elicits a Progressive Loss of Fe-S Protein Expression.” That is sort of trivial, has been in the center of previous studies by other labs, and is used as a control here (a,b). The new part c of the Cys deprivation is of interest here, and should make the headline ‘Cys deprivation leads to cytosolic and nuclear FeS protein defects’ (opposite to what is written in the text).

1. Good and necessary addition. Resolved.
2. I accept the experimental limitations. Resolved.

3. The inclusion of the additional data is helpful for the reader, and (at least partially) resolves this issue.

4. Resolved.

5. The inclusion of a schematic representation (Fig. 3f) of the experimental design is very helpful. But now I am even more confused by the presentation of the data: after 48h of NFS1 or ISCU depletion (i.e. time point 0), the FeS activities should be much lower than WT. This is not evident from the panels 3g-i. Did the authors normalize the (lower?) activities to 100%? Possibly the (further) decline of the activities in the ISC depletions indicates further decline of the NFS1 and ISCU proteins. I suggest to point that out to the reader. It may be better present the original and not normalized data. The result of the Figure is clear: In comparison to ISC depletions, the Cys2 deprivation has no (or even a positive) effect.

6. Resolved. Aco2 activity looks more convincing than respiratory complexes.

7. The mitochondrial fraction of the CHAC1 protein is now claimed to be in the intermembrane space (IMS) based on its protease sensitivity. At this stage of analysis, this is only a suggestion, since the argument is based on mitochondria with heavily damaged outer membranes (IMS is accessible to PK without swelling). While mitochondrial opening is often unavoidable during organelle isolation, I see that also ACO2, NDUFS1 and CS are quite sensitive to PK (does NNT expose a domain to the IMS?). I am not sure whether quantitation of all these blots (plus the repeats) would reveal a clear distinction in protease sensitivity between matrix and IMS proteins. Without an obvious mito targeting information in CHAC1, I strongly suggest to tone down the statement on its exact sub-mito localization. What seems to be clear: A fraction of the protein is inside mitochondria and not simply sticking outside, but its precise sub-mito localization needs to be worked out (which may not be in the scope of this study).

8. Resolved.

9. I am not really convinced by the explanations. But it is in the authors' responsibility to maintain the claim.

10. The criticized Fig. 5 was changed by enlarge. The criticized data on CHAC1 depletion have been moved to Extended Data Fig. 6c-f, and replaced by NFS1 depletion data. I maintain that these results are not convincing. The new data support the notion that GSH is converted by CHAC1 to Cys which is used by NFS1 (and presumably proteins synthesis). If NFS1 is depleted, GSH is less (or not consumed). Altogether, the issue is resolved.

What generally made this part of the manuscript hard to re-review, is that the authors were not correctly highlighting what was changed in the revised version. The highlighted text in lines 301-317 is in fact unchanged, but the headline is new (with a new interpretation statement). Very confusing (a Word 'Compare mode' version would be more suitable).

11. This result (i.e. no Cys no GSH, but FeS and protein biosynthesis are fine) is really surprising. But cytosolic FeS proteins are decreased (see above), meaning they are the most sensitive part of GSH depletion (as found earlier; see above).

12. It is a good idea to remove this data from the manuscript. This was too preliminary and may have been an overinterpretation.

Minor additional points:

- In Ext.Fig. 5 b-d the (green) colors can hardly be discriminated. More distinct colors are advisable. I first thought, two bars are missing.

- Line 285: complimentary temporal  complementary temporal (what exactly is meant by this??)

- It would be helpful to explicitly state by which strategy mito-CHAC1 was targeted to mitochondria (which presequence?). At present, the reader is forced to go to Addgene and dig through a forest...

Reviewer #3 (Remarks to the Author):

The authors have adequately addressed all concerns from this reviewer.

We thank the reviewer again for their continued effort in reviewing our manuscript and are pleased to hear that we have sufficiently addressed their remaining concerns.

Reviewer #4 (Remarks to the Author):

I read the new manuscript and the rebuttal letter, particularly wrt my concerns. The authors have replied to all of my concerns, and in many cases (see below), the issues are resolved. In a few cases, the issues are still open, but the final experimental addressing may be part of another work. For some cases, however, the authors interpret their (new) data differently from what I see from the presented data. This is a major remaining concern that, in my view, needs to be addressed. For bringing clarity into the entire story, I suggest that the authors draw a model for their views that consistently covers all the observations presented in the manuscript. I am convinced that this would help the authors (and the reviewer as well as the later readers) to judge whether all parts of the findings fit into that model. My specific comments on the rebuttal letter and the revised version of this manuscript are given below (I refer to the sub-points used by the authors in the rebuttal letter):

We thank the reviewer for the additional effort put forth in reviewing our manuscript. We have revised the manuscript to address their concerns where appropriate and have included a more comprehensive model figure (Figure 6h) summarizing the broad findings of the study. Please find below specific responses to each of the critiques with references to where the corresponding edits appear in the revised manuscript.

Summary Figure: Mitochondrial Fe-S cluster synthesis is more resistant to sulfur than iron restriction due to the mobilization of cysteine stored in mitochondrial glutathione by CHAC1. This CHAC1 activity supports persistent respiratory function and the induction of ferroptosis under prolonged cysteine deprivation.

ii) The authors immunodetected several cytoplasmic and nuclear FeS proteins. As I was suspecting, many of these FeS proteins showed a marked decline in stability (amount) over the time course. Particularly, DPYD and NTHL (I look at the major band) levels go down, while the data for other proteins are not clear without quantitation of band intensities (and the repeats). I do not understand why the authors interpret this data set as “Fe-S cluster synthesis is sustained under the period of cystine starvation”. The opposite is what I see. This mis-interpretation of the data is more than confusing (mis-leading the reader). Further, I would expect that this FeS protein decline would be reflected in a decrease in enzyme activities, eg cytosolic ACO1. This data is not provided, and is needed here. Hence, my criticism is not resolved, because the interpretation of the story is incorrect, in my view. What the new result, however, means is that GSH depletion (due to its degradation upon Cys starvation) leads to cytosolic FeS defects. This exactly mimics what was seen earlier in yeast (PMID 12011041, 21478822). The title of the manuscript is therefore no longer correct, because biogenesis of cytoplasmic FeS proteins is a key function of mitochondria.

We have revised the manuscript text to note the variability in extramitochondrial Fe-S protein stability under cystine starvation (**Lines 211-214**). Further, we have included data reflecting ACO1 activity in response to the various insults to Fe-S cluster synthesis (as illustrated in Figure 3f), which shows that ACO1 activity is far more resistant to cystine starvation than the other treatments (**Extended Data Figure 2d**). We have also revised the title of the manuscript to read “Mitochondrial Respiratory Function is Preserved Under Cysteine Starvation via Glutathione Catabolism in NSCLC” to specifically reflect the major finding of our study.

ACO1 Activity is More Resistant to Cysteine than Iron Limitation. Time course analysis of ACO1 activity in response to cysteine starvation, iron chelation, or shRNA-mediated knockdown of NFS1 or ISCU.

Though we agree that these changes have improved the manuscript, we strongly disagree with the reviewer’s critique that the broad interpretation of our data is incorrect. We contend that our findings that mitochondrial respiration, mitochondrial Fe-S protein function and stability are all maintained under 24h of cystine starvation is clear evidence of sustained mitochondrial Fe-S cluster synthesis. The variability in extramitochondrial Fe-S protein stability does not in itself indicate a defect in mitochondrial Fe-S cluster synthesis, especially when we observe no defect in relevant mitochondrial biology. While it has been described that nascent Fe-S clusters are exported from the mitochondria to support the function of cytosolic/nuclear Fe-S proteins (PMID: 10406803), there is also evidence to indicate *de novo* synthesis of Fe-S clusters within the cytosol (PMID: 16527810; 29309586). Therefore, our observation of increased sensitivity of certain extramitochondrial Fe-S proteins (e.g., DPYD, NTHL1) to cystine starvation could indicate a number of possibilities, including i.) a defect in cytosolic cluster synthesis due to the absence of cysteine, ii.) regulation of the export of mitochondrially-derived clusters, iii.) hierarchal regulation of cluster trafficking to support the function of certain proteins at the expense of others when the pool of clusters is limiting, and iv.) increased sensitivity of the Fe-S clusters within these proteins to ROS-mediated oxidation, as we show in Figure 2 that cytosolic ROS and lipid oxidation do increase upon cysteine starvation. We have expanded the discussion (**Lines 448-459**) to raise these possibilities, however, it is far beyond the scope of this study to elucidate the operative mechanism. While this is

incredibly interesting biology, this manuscript is focused on how CHAC1 acts to support the unexpected retention of mitochondrial respiratory function under cysteine limitation in NSCLC cells.

Further note: The headline of ExFig. 2 does not really express what the Figure shows. “Disruption of Fe-S Cluster Synthesis Elicits a Progressive Loss of Fe-S Protein Expression.” That is sort of trivial, has been in the center of previous studies by other labs, and is used as a control here (a,b). The new part c of the Cys deprivation is of interest here, and should make the headline ‘Cys deprivation leads to cytosolic and nuclear FeS protein defects’ (opposite to what is written in the text).

Again, the focus of this work was on the mechanism by which mitochondrial cysteine metabolism (e.g. Fe-S cluster synthesis) persists in NSCLC when cells are starved of an extracellular source. While it is interesting that some extramitochondrial proteins demonstrate reduced expression under cysteine deprivation we disagree with the blanket claim that “Cys deprivation leads to cytosolic and nuclear FeS protein defects” based on the data in Extended Figure 2. Therefore, we have changed the title of Extended Data Figure 2 to read “Extramitochondrial Fe-S Proteins Exhibit Varied Stability in Response to Cystine Starvation”. We feel that this is an accurate representation of the data and does not unduly speculate on the reason for the varied effect, which again is beyond the scope of this work. Furthermore, we have also revised the associated headline within the manuscript text to read “Mitochondrial Fe-S Cluster Synthesis is Sustained Under Cystine Deprivation” (**Line 159**). This is a clear and specific reflection of our data and removes any ambiguity in the presentation of our findings as it relates to the cytosolic pathway.

5. The inclusion of a schematic representation (Fig. 3f) of the experimental design is very helpful. But now I am even more confused by the presentation of the data: after 48h of NFS1 or ISCU depletion (i.e. time point 0), the FeS activities should be much lower than WT. This is not evident from the panels 3g-i. Did the authors normalize the (lower?) activities to 100%? Possibly the (further) decline of the activities in the ISC depletions indicates further decline of the NFS1 and ISCU proteins. I suggest to point that out to the reader. It may be better present the original and not normalized data. The result of the Figure is clear: In comparison to ISC depletions, the Cys2 deprivation has no (or even a positive) effect.

For clarity of the overall figure panel, we have kept the existing formatting, however, we have revised the figure legends for **Figures 3g-i** and **Extended Data Figures 2d** to make it clear that the data were normalized to the activity of each protein at time point 0h for each group.

7. The mitochondrial fraction of the CHAC1 protein is now claimed to be in the intermembrane space (IMS) based on its protease sensitivity. At this stage of analysis, this is only a suggestion, since the argument is based on mitochondria with heavily damaged outer membranes (IMS is accessible to PK without swelling). While mitochondrial opening is often unavoidable during organelle isolation, I see that also ACO2, NDUFS1 and CS are quite sensitive to PK (does NNT expose a domain to the IMS?). I am not sure whether quantitation of all these blots (plus the repeats) would reveal a clear distinction in protease sensitivity between matrix and IMS proteins. Without an obvious mito targeting information in CHAC1, I strongly suggest to tone down the statement on its exact sub-mito localization. What seems to be clear: A fraction of the protein is inside mitochondria and not simply sticking outside, but its precise sub-mito localization needs to be worked out (which may not be in the scope of this study).

We agree that determining the precise sub-mitochondrial localization is beyond the scope of this study and that the totality of our data is sufficient to claim that mitochondrial CHAC1 is not simply associated with the cytosolic face of the outer mitochondrial membrane. Still, to avoid an overinterpretation of our existing data, we have revised the relevant manuscript text (**Lines 273-275, 333-334, 440**) to tone down the certainty with which we suggest that CHAC1 is associated with the IMS.

Minor additional points:

- In Ext.Fig. 5 b-d the (green) colors can hardly be discriminated. More distinct colors are advisable. I first thought, two bars are missing.

For consistency purposes we have retained a green color scheme for panels relating to the manipulation of CHAC1 (**Figures 5e and f, and Extended Data Figures 4 and 5**), however, we have incorporated a darker green color for bars representing shCHAC1 #2 to better differentiate it from the other two hairpins that we employed.

- Line 285: complimentary temporal  complementary temporal (what exactly is meant by this??)
We have edited the text to remedy the spelling mistake. The text in question is referencing our time-course based (0-96h) analyses of Fe-S protein function that complement the analyses performed at 20h of cystine starvation (**Extended Data Figure 4**).

- It would be helpful to explicitly state by which strategy mito-CHAC1 was targeted to mitochondria (which presequence?). At present, the reader is forced to go to Addgene and dig through a forest...

We apologize for the omission; we have added detail to the methods section (**Lines 524-527**) describing how we employed the leader sequence of ornithine transcarbamylase to target CHAC1 (MitoCHAC1) to the mitochondria.

REVIEWERS' COMMENTS

Reviewer #4 (Remarks to the Author):

I read the new manuscript and the rebuttal letter, particularly wrt my concerns. The authors have replied to all of my concerns, and in many cases (see below), the issues are resolved. In a few cases, the issues are still open, but the final experimental addressing may be part of another work. For some cases, however, the authors interpret their (new) data differently from what I see from the presented data. This is a major remaining concern that, in my view, needs to be addressed. For bringing clarity into the entire story, I suggest that the authors draw a model for their views that consistently covers all the observations presented in the manuscript. I am convinced that this would help the authors (and the reviewer as well as the later readers) to judge whether all parts of the findings fit into that model. My specific comments on the rebuttal letter and the revised version of this manuscript are given below (I refer to the sub-points used by the authors in the rebuttal letter):

We thank the reviewer for the additional effort put forth in reviewing our manuscript. We have revised the manuscript to address their concerns where appropriate and have included a more comprehensive model figure (Figure 6h) summarizing the broad findings of the study. Please find below specific responses to each of the critiques with references to where the corresponding edits appear in the revised manuscript.

Summary Figure: Mitochondrial Fe-S cluster synthesis is more resistant to sulfur than iron restriction due to the mobilization of cysteine stored in mitochondrial glutathione by CHAC1. This CHAC1 activity supports persistent respiratory function and the induction of ferroptosis under prolonged cysteine deprivation.

RE: This model figure surely helps the readers in understanding the message of the paper. However, it lacks a number of the important findings of this manuscript, and, in the interest of the authors, I would add these aspects. What they for instance see is a maintenance of mitochondrial, yet a defect in cytosolic FeS protein assembly (see point ii below) as a result of glutathione depletion by CHAC1, mediated by its conversion to Cys. As previously mentioned, this finding exactly fits the widely accepted model of the FeS protein biogenesis field in that GSH (and mitochondria) is needed primarily for cytosolic FeS biosynthesis (see my previous comment (PMID 12011041, 21478822) and models in doi:10.1039/c7mt00269f; 10.1016/j.bbamcr.2020.118863). This point is easy to add. What is also missing is an explanation of the abbreviations (eg PLO and PLOO, etc.).

ii) The authors immunodetected several cytoplasmic and nuclear FeS proteins. As I was suspecting, many of these FeS proteins showed a marked decline in stability (amount) over the time course. Particularly, DPYD and NTHL (I look at the major band) levels go down, while the data for other proteins are not clear

without quantitation of band intensities (and the repeats). I do not understand why the authors interpret this data set as “Fe-S cluster synthesis is sustained under the period of cystine starvation”. The opposite is what I see. This mis-interpretation of the data is more than confusing (mis- leading the reader). Further, I would expect that this FeS protein decline would be reflected in a decrease in enzyme activities, eg cytosolic ACO1. This data is not provided, and is needed here. Hence, my criticism is not resolved, because the interpretation of the story is incorrect, in my view. What the new result, however, means is that GSH depletion (due to its degradation upon Cys starvation) leads to cytosolic FeS defects. This exactly mimics what was seen earlier in yeast (PMID 12011041, 21478822). The title of the manuscript is therefore no longer correct, because biogenesis of cytoplasmic FeS proteins is a key function of mitochondria.

We have revised the manuscript text to note the variability in extramitochondrial Fe-S protein stability under cystine starvation (Lines 211-214). Further, we have included data reflecting ACO1 activity in response to the various insults to Fe-S cluster synthesis (as illustrated in Figure 3f), which shows that ACO1 activity is far more resistant to cystine starvation than the other treatments (Extended Data Figure 2d). We have also revised the title of the manuscript to read “Mitochondrial Respiratory Function is Preserved Under Cysteine Starvation via Glutathione Catabolism in NSCLC” to specifically reflect the major finding of our study.

RE: The new result of Suppl. Fig. 2d aligns well with the blot in 2c for H1299. In comparison to mitoACO2 (Fig. 2g), also this activity goes down, and for the H2009 cells one would expect an even stronger decline of the activity (unfortunately not analyzed). Considering the trends seen in the time courses observed for other cytosol or nucleus FeS proteins, an even stronger decrease is likely for extended time periods (eg 36 h). Overall, this fits with the model I suggested above that mitochondria and GSH are needed for cytosolic FeS protein biosynthesis.

Though we agree that these changes have improved the manuscript, we strongly disagree with the reviewer’s critique that the broad interpretation of our data is incorrect. We contend that our findings that mitochondrial respiration, mitochondrial Fe-S protein function and stability are all maintained under 24h of cystine starvation is clear evidence of sustained mitochondrial Fe-S cluster synthesis.

RE: The authors misunderstood. I am not saying that mitochondrial activity is hampered, but their role (and that of GSH) in cytosolic FeS biosynthesis is.

The variability in extramitochondrial Fe-S protein stability does not in itself indicate a defect in mitochondrial Fe-S cluster synthesis, especially when we observe no defect in relevant mitochondrial biology. While it has been described that nascent Fe-S clusters are exported from the mitochondria to support the function of cytosolic/nuclear Fe-S proteins (PMID: 10406803), there is also evidence to indicate de novo synthesis of Fe-S clusters within the cytosol (PMID: 16527810; 29309586). Therefore, our observation of increased sensitivity of certain extramitochondrial Fe-S proteins (e.g., DPYD, NTHL1) to cystine starvation could indicate a number of possibilities, including i.) a defect in cytosolic cluster synthesis due to the absence of cysteine, ii.) regulation of the export of mitochondrially-derived clusters, iii.) hierarchal regulation of cluster trafficking to support the function of certain proteins at the expense of others when the pool of clusters is limiting, and iv.) increased sensitivity of the Fe-S clusters within

these proteins to ROS- mediated oxidation, as we show in Figure 2 that cytosolic ROS and lipid oxidation do increase upon cysteine starvation. We have expanded the discussion (Lines 448-459) to raise these possibilities, however, it is far beyond the scope of this study to elucidate the operative mechanism. While this is incredibly interesting biology, this manuscript is focused on how CHAC1 acts to support the unexpected retention of mitochondrial respiratory function under cysteine limitation in NSCLC cells.

RE: As said above, the observations of the authors nicely support the well-accepted model of FeS protein biosynthesis and the role of mitochondria for cytosolic FeS protein assembly. Of course, the authors are free to offer other possibilities. However, numerous *in vivo* studies (including this work 10.1074/jbc.M112.418889) have shown that the mitochondrial ISC machinery is required for cytosolic FeS assembly.

Further note: The headline of ExFig. 2 does not really express what the Figure shows. "Disruption of Fe- S Cluster Synthesis Elicits a Progressive Loss of Fe-S Protein Expression." That is sort of trivial, has been in the center of previous studies by other labs, and is used as a control here (a,b). The new part c of the Cys deprivation is of interest here, and should make the headline 'Cys deprivation leads to cytosolic and nuclear FeS protein defects' (opposite to what is written in the text).

Again, the focus of this work was on the mechanism by which mitochondrial cysteine metabolism (e.g. Fe-S cluster synthesis) persists in NSCLC when cells are starved of an extracellular source. While it is interesting that some extramitochondrial proteins demonstrate reduced expression under cysteine deprivation we disagree with the blanket claim that "Cys deprivation leads to cytosolic and nuclear FeS protein defects" based on the data in Extended Figure 2. Therefore, we have changed the title of Extended Data Figure 2 to read "Extramitochondrial Fe-S Proteins Exhibit Varied Stability in Response to Cystine Starvation". We feel that this is an accurate representation of the data and does not unduly speculate on the reason for the varied effect, which again is beyond the scope of this work. Furthermore, we have also revised the associated headline within the manuscript text to read "Mitochondrial Fe-S Cluster Synthesis is Sustained Under Cystine Deprivation" (Line 159). This is a clear and specific reflection of our data and removes any ambiguity in the presentation of our findings as it relates to the cytosolic pathway.

RE: Resolved.

5. The inclusion of a schematic representation (Fig. 3f) of the experimental design is very helpful. But now I am even more confused by the presentation of the data: after 48h of NFS1 or ISCU depletion (i.e. time point 0), the FeS activities should be much lower than WT. This is not evident from the panels 3g-i. Did the authors normalize the (lower?) activities to 100%? Possibly the (further) decline of the activities in the ISC depletions indicates further decline of the NFS1 and ISCU proteins. I suggest to point that out to the reader. It may be better present the original and not normalized data. The result of the Figure is clear: In comparison to ISC depletions, the Cys2 deprivation has no (or even a positive) effect.

RE: For clarity of the overall figure panel, we have kept the existing formatting, however, we have revised the figure legends for Figures 3g-i and Extended Data Figures 2d to make it clear that the data were normalized to the activity of each protein at time point 0h for each group.

RE: Resolved.

7. The mitochondrial fraction of the CHAC1 protein is now claimed to be in the intermembrane space (IMS) based on its protease sensitivity. At this stage of analysis, this is only a suggestion, since the argument is based on mitochondria with heavily damaged outer membranes (IMS is accessible to PK without swelling). While mitochondrial opening is often unavoidable during organelle isolation, I see that also ACO2, NDUFS1 and CS are quite sensitive to PK (does NNT expose a domain to the IMS?). I am not sure whether quantitation of all these blots (plus the repeats) would reveal a clear distinction in protease sensitivity between matrix and IMS proteins. Without an obvious mito targeting information in CHAC1, I strongly suggest to tone down the statement on its exact sub-mito localization. What seems to be clear: A fraction of the protein is inside mitochondria and not simply sticking outside, but its precise sub-mito localization needs to be worked out (which may not be in the scope of this study).

We agree that determining the precise sub-mitochondrial localization is beyond the scope of this study and that the totality of our data is sufficient to claim that mitochondrial CHAC1 is not simply associated with the cytosolic face of the outer mitochondrial membrane. Still, to avoid an overinterpretation of our existing data, we have revised the relevant manuscript text (Lines 273-275, 333-334, 440) to tone down the certainty with which we suggest that CHAC1 is associated with the IMS.

RE: Resolved.

Minor additional points:

- In Ext.Fig. 5 b-d the (green) colors can hardly be discriminated. More distinct colors are advisable. I first thought, two bars are missing.

For consistency purposes we have retained a green color scheme for panels relating to the manipulation of CHAC1 (Figures 5e and f, and Extended Data Figures 4 and 5), however, we have incorporated a darker green color for bars representing shCHAC1 #2 to better differentiate it from the other two hairpins that we employed.

RE: Resolved

- Line 285: complimentary temporal  complementary temporal (what exactly is meant by this??)
We have edited the text to remedy the spelling mistake. The text in question is referencing our time-course based (0-96h) analyses of Fe-S protein function that complement the analyses performed at 20h of cystine starvation (Extended Data Figure 4).

RE: Resolved

- It would be helpful to explicitly state by which strategy mito-CHAC1 was targeted to mitochondria (which presequence?). At present, the reader is forced to go to Addgene and dig through a forest...

We apologize for the omission; we have added detail to the methods section (Lines 524-527) describing how we employed the leader sequence of ornithine transcarbamylase to target CHAC1 (MitoCHAC1) to the mitochondria.

RE: Resolved. The authors may wish to add a comment that this leads to matrix localization (rather than the presumed IMS localization of the native enzyme).

Reviewer #4 (Remarks to the Author):

1. Summary Figure: Mitochondrial Fe-S cluster synthesis is more resistant to sulfur than iron restriction due to the mobilization of cysteine stored in mitochondrial glutathione by CHAC1. This CHAC1 activity supports persistent respiratory function and the induction of ferroptosis under prolonged cysteine deprivation.

RE: This model figure surely helps the readers in understanding the message of the paper. However, it lacks a number of the important findings of this manuscript, and, in the interest of the authors, I would add these aspects. What they for instance see is a maintenance of mitochondrial, yet a defect in cytosolic FeS protein assembly (see point ii below) as a result of glutathione depletion by CHAC1, mediated by its conversion to Cys. As previously mentioned, this finding exactly fits the widely accepted model of the FeS protein biogenesis field in that GSH (and mitochondria) is needed primarily for cytosolic FeS biosynthesis (see my previous comment (PMID 12011041, 21478822) and models in doi:10.1039/c7mt00269f; 10.1016/j.bbamcr.2020.118863). This point is easy to add. What is also missing is an explanation of the abbreviations (eg PLO and PLOO, etc.).

We have modified the model figure (**Figure 6h**) to specifically show that mitochondrial Fe-S cluster synthesis is more resistant to sulfur than iron restriction and that cysteine starvation does inhibit the cytosolic iron sulfur cluster assembly system (CIA). Since we did not experimentally demonstrate that this was a specific consequence of CHAC1-mediated GSH degradation, we refrained from depicting this hypothesis. However, we did amend the discussion to highlight that our findings are consistent with previous work demonstrating the need for GSH in cytosolic cluster assembly as you have suggested (**Lines 447-451**).

Thank you for pointing out this omission, we have now defined the membrane phospholipid alkoxyl (PLO•) and peroxy (PLOO•) radicals in the figure legend.

Summary Figure: Mitochondrial Fe-S cluster synthesis is more resistant to sulfur (Cys) than iron restriction due to CHAC1 catabolism of mitochondrial GSH. This contrasts with the cytosolic iron sulfur cluster assembly system (CIA), which exhibits sensitivity to Cys starvation. The persistence of respiratory function under Cys limitation potentiates the iron-mediated generation of membrane phospholipid alkoxyl (PLO•) and peroxy (PLOO•) radicals that mediate ferroptosis.

2. The variability in extramitochondrial Fe-S protein stability does not in itself indicate a defect in mitochondrial Fe-S cluster synthesis, especially when we observe no defect in relevant mitochondrial biology. While it has been described that nascent Fe-S clusters are exported from the mitochondria to support the function of cytosolic/nuclear Fe-S proteins (PMID: 10406803), there is also evidence to indicate de novo synthesis of Fe-S clusters within the cytosol (PMID: 16527810; 29309586). Therefore, our observation of increased sensitivity of certain extramitochondrial Fe-S proteins (e.g., DPYD, NTHL1) to cystine starvation could indicate a number of possibilities, including i.) a defect in cytosolic cluster synthesis due to the absence of cysteine, ii.) regulation of the export of mitochondrially-derived clusters, iii.) hierarchical regulation of cluster trafficking to support the function of certain proteins at the expense of others when the pool of clusters is limiting, and iv.) increased sensitivity of the Fe-S clusters within these proteins to ROS-mediated oxidation, as we show in Figure 2 that cytosolic ROS and lipid oxidation do increase upon cysteine starvation. We have expanded the discussion (Lines 448-459) to raise these possibilities, however, it is far beyond the scope of this study to elucidate the operative mechanism. While this is incredibly interesting biology, this manuscript is focused on how CHAC1 acts to support the unexpected retention of mitochondrial respiratory function under cysteine limitation in NSCLC cells.

RE: As said above, the observations of the authors nicely support the well-accepted model of FeS protein biosynthesis and the role of mitochondria for cytosolic FeS protein assembly. Of course, the authors are free to offer other possibilities. However, numerous *in vivo* studies (including this work 10.1074/jbc.M112.418889) have shown that the mitochondrial ISC machinery is required for cytosolic FeS assembly.

Based on the totality of our data, we are not comfortable making the conclusion that the effect of cysteine starvation on the CIA is definitively related to the mitochondria. Our interpretation of the state of the field is that there is a spectrum of opinions regarding the nature and complexity of cytosolic Fe-S cluster synthesis/maturation in mammalian systems that range from being explicitly dependent on the export of mitochondrially derived clusters (PMID: 28615445) to a fully compartmentalized system within the cytosol (PMID: 30923992). In fact, the senior author of the reference you cite here is an outspoken advocate for the existence of a distinct cytosolic pathway of Fe-S cluster synthesis (PMID: 11060020; 30923992), and has raised skepticism over the necessity of the mitochondria in the generation of mammalian cytosolic clusters due to those championed studies being performed largely in yeast (PMID: 32311335). The true biology likely lies somewhere in-between, and our data do not necessarily support or contradict either mechanism. We have amended the discussion to state that we did not distinguish the effect of cysteine starvation between these two major models of the CIA, but again highlighted how the observed effects on cytosolic cluster stability are at least in part related to the established role of GSH in the maturation of cytosolic clusters (**Lines 447-451**).

3. We apologize for the omission; we have added detail to the methods section (Lines 524-527) describing how we employed the leader sequence of ornithine transcarbamylase to target CHAC1 (MitoCHAC1) to the mitochondria.

RE: Resolved. The authors may wish to add a comment that this leads to matrix localization (rather than the presumed IMS localization of the native enzyme).

We have clarified the methods section to state that this leads to mitochondrial matrix localization (**Lines 524-525**).